# Gut microbial trimethylamine is elevated in alcohol-associated hepatitis and contributes to ethanol-induced liver injury in mice

Robert N Helsley[1,2,3†], Tatsunori Miyata[4†], Anagha Kadam[1,2†],
Venkateshwari Varadharajan[1,2], Naseer Sangwan[1,2], Emily C Huang[4],
Rakhee Banerjee[1,2], Amanda L Brown[1,2], Kevin K Fung[1,2], William J Massey[1,2],
Chase Neumann[1,2], Danny Orabi[1,2], Lucas J Osborn[1,2], Rebecca C Schugar[1,2],
Megan R McMullen[4], Annette Bellar[4], Kyle L Poulsen[4], Adam Kim[4], Vai Pathak[5],
Marko Mrdjen[1,2,4], James T Anderson[1,2], Belinda Willard[1,2], Craig J McClain[6],
Mack Mitchell[7], Arthur J McCullough[2,4], Svetlana Radaeva[8], Bruce Barton[9],
Gyongyi Szabo[10], Srinivasan Dasarathy[2,4], Jose Carlos Garcia-Garcia[11],
Daniel M Rotroff[5], Daniela S Allende[12], Zeneng Wang[1,2], Stanley L Hazen[1,2,13],
Laura E Nagy[2,4], Jonathan Mark Brown[1,2]*

[1]Department of Cardiovascular and Metabolic Sciences, Lerner Research Institute of the Cleveland Clinic, Cleveland, United States; [2]Center for Microbiome and Human Health, Lerner Research Institute, Cleveland Clinic, Cleveland, United States; [3]Department of Pediatrics, Division of Pediatric Gastroenterology, Hepatology, and Nutrition, College of Medicine, University of Kentucky, Lexington, United States; [4]Department of Inflammation and Immunity, Lerner Research Institute, Cleveland Clinic, Cleveland, United States; [5]Department of Quantitative Health Sciences, Lerner Research Institute, Cleveland Clinic, Cleveland, United States; [6]Department of Medicine, University of Louisville, Louisville, United States; [7]Department of Internal Medicine, University of Texas Southwestern Medical Center, Dallas, United States; [8]National Institute on Alcohol Abuse and Alcoholism, Bethesda, United States; [9]Department of Population and Quantitative Health Sciences, University of Massachusetts Medical School, Worcester, United States; [10]Department of Medicine, Beth Israel Deaconess Medical Center and Harvard Medical School, Boston, United States; [11]Life Sciences Transformative Platform Technologies, Procter & Gamble, Cincinnati, United States; [12]Department of Anatomical Pathology, Cleveland Clinic, Cleveland, United States; [13]Department of Cardiovascular Medicine, Heart and Vascular and Thoracic Institute, Cleveland Clinic, Cleveland, United States

*For correspondence:
brownm5@ccf.org

†These authors contributed equally to this work

**Abstract** There is mounting evidence that microbes residing in the human intestine contribute to diverse alcohol-associated liver diseases (ALD) including the most deadly form known as alcohol-associated hepatitis (AH). However, mechanisms by which gut microbes synergize with excessive alcohol intake to promote liver injury are poorly understood. Furthermore, whether drugs that selectively target gut microbial metabolism can improve ALD has never been tested. We used liquid chromatography tandem mass spectrometry to quantify the levels of microbe and host choline co-metabolites in healthy controls and AH patients, finding elevated levels of the microbial metabolite trimethylamine (TMA) in AH. In subsequent studies, we treated mice with non-lethal bacterial

choline TMA lyase (CutC/D) inhibitors to blunt gut microbe-dependent production of TMA in the context of chronic ethanol administration. Indices of liver injury were quantified by complementary RNA sequencing, biochemical, and histological approaches. In addition, we examined the impact of ethanol consumption and TMA lyase inhibition on gut microbiome structure via 16S rRNA sequencing. We show the gut microbial choline metabolite TMA is elevated in AH patients and correlates with reduced hepatic expression of the TMA oxygenase flavin-containing monooxygenase 3 (FMO3). Provocatively, we find that small molecule inhibition of gut microbial CutC/D activity protects mice from ethanol-induced liver injury. CutC/D inhibitor-driven improvement in ethanol-induced liver injury is associated with distinct reorganization of the gut microbiome and host liver transcriptome. The microbial metabolite TMA is elevated in patients with AH, and inhibition of TMA production from gut microbes can protect mice from ethanol-induced liver injury.

## Editor's evaluation

This paper aims to understand the mechanisms by which gut microbes synergize with excessive alcohol intake to cause liver injury, and whether drugs that selectively target gut microbial metabolism can improve alcohol-associated liver disease (ALD). The authors used liquid chromatography tandem mass spectrometry to quantify the levels of microbe and host choline co-metabolites in controls and patients with alcohol-associated hepatitis (AH). They also treated mice with bacterial choline trimethylamine (TMA) lyase inhibitors to reduce gut microbe-dependent TMA production, followed by measurement of Indices of liver injury. They showed that gut microbial choline metabolite TMA is increased in AH patients, which correlates with reduced liver expression of the TMA oxygenase Flavin-containing monooxygenase 3 (FMO3). They also show that inhibition of gut microbial CutC/D activity protects from ethanol-induced liver injury in mouse models, which was associated with reorganization of the gut microbiome and host liver transcriptome. The authors conclude that microbial TMA is elevated in patients with AH, and inhibition of TMA production by gut microbes protects against ethanol-induced liver injury.

## Introduction

Alcohol-associated liver disease (ALD) includes a spectrum of liver pathologies including steatosis, fibrosis, cirrhosis, and the most severe manifestation known as alcohol-associated hepatitis (AH). Shortly after diagnosis AH patients die at a staggering rate of 40–50% (*Masarone et al., 2016*; *Kochanek et al., 2017*). Despite many attempts, an effective therapy for this deadly disease has been elusive. Similar to other components of the spectrum of ALD, AH has consistently been linked to reorganization of the gut microbiome and dysregulation of microbe-host interactions (*Chen et al., 2011*; *Yan et al., 2011*; *Mutlu et al., 2009*; *Mutlu et al., 2012*; *Tripathi et al., 2018*; *Ciocan et al., 2018*; *Llopis et al., 2016*; *Duan et al., 2019*; *Smirnova et al., 2020*; *Gao et al., 2019*; *Puri et al., 2018*; *Lang and Schnabl, 2020*). It is well appreciated that chronic alcohol use can elicit structural alterations in the gut barrier, allowing either live bacteria themselves or microbe-associated molecule patterns (MAMPs), such as lipopolysaccharide (LPS), to enter the portal circulation where they can directly engage pattern recognition receptors (PRRs) such as Toll-like receptors (TLRs) or NOD-like receptors (NLRP3, NLRP6, etc.) to promote hepatic inflammation and tissue injury (*Wilkinson et al., 1974*; *Tarao et al., 1979*; *Uesugi et al., 2001*; *Paik et al., 2003*; *DeSantis et al., 2013*; *Knorr et al., 2020*). In addition to MAMP-PRR interactions, gut microbes can act as a collective endocrine organ, producing a vast array of small molecules, proteins, and lipid metabolites that can engage dedicated host receptor systems to also impact liver disease progression (*Brown and Hazen, 2015*). Collectively, these MAMP-PRR and microbial metabolite-host receptor interactions converge to promote ALD and many other diseases of uncontrolled inflammation (*Brown and Hazen, 2015*; *Gilbert et al., 2018*).

Although there is now clear evidence that microbe-host interactions play a key role in liver disease progression (*Chen et al., 2011*; *Yan et al., 2011*; *Mutlu et al., 2009*; *Mutlu et al., 2012*; *Tripathi et al., 2018*; *Ciocan et al., 2018*; *Llopis et al., 2016*; *Duan et al., 2019*; *Smirnova et al., 2020*; *Gao et al., 2019*; *Puri et al., 2018*; *Lang and Schnabl, 2020*; *Wilkinson et al., 1974*; *Tarao et al., 1979*; *Uesugi et al., 2001*; *Paik et al., 2003*; *DeSantis et al., 2013*; *Knorr et al., 2020*; *Brown and Hazen, 2015*; *Gilbert et al., 2018*), ALD drug discovery to this point has focused primarily on targets encoded by the

human genome. Our knowledge is rapidly expanding as to how microbes intersect with ALD progression, including cataloging microbial genomes. We also now understand the repertoire of MAMPs gut microbes harbor as well as the vast array of metabolites that they produce in both patients with ALD and animal models of ethanol-induced liver injury (*Chen et al., 2011*; *Yan et al., 2011*; *Mutlu et al., 2009*; *Mutlu et al., 2012*; *Tripathi et al., 2018*; *Ciocan et al., 2018*; *Llopis et al., 2016*; *Duan et al., 2019*; *Smirnova et al., 2020*; *Gao et al., 2019*; *Puri et al., 2018*; *Lang and Schnabl, 2020*; *Wilkinson et al., 1974*; *Tarao et al., 1979*; *Uesugi et al., 2001*; *Paik et al., 2003*; *DeSantis et al., 2013*; *Knorr et al., 2020*; *Brown and Hazen, 2015*; *Gilbert et al., 2018*). However, there are very few examples of where this information has been leveraged into safe and effective therapeutic strategies. In general, the microbiome-targeted therapeutic field has primarily focused on either anti-, pre-, or pro-biotic approaches, yet these microbial community-restructuring approaches have resulted in very modest or non-significant effects in clinical studies of liver disease (*Kwak et al., 2014*; *Asgharian et al., 2020*; *Reijnders et al., 2016*; *Madjd et al., 2016*). As an alternative microbiome-targeted approach, we and others have begun developing non-lethal selective small molecule inhibitors of bacterial enzymes with the intention of reducing levels of disease-associated microbial metabolites with mechanistic rationale for contribution to disease pathogenesis (*Roberts et al., 2018*; *Wang et al., 2015*; *Gupta et al., 2020*; *Organ et al., 2020*; *Orman et al., 2019*). In fact, we have recently shown that small molecule inhibition of the gut microbial transformation of choline into trimethylamine (TMA), the initial and rate-limiting step in the generation of the cardiovascular disease (CVD)-associated metabolite trimethylamine N-oxide (TMAO), can significantly reduce disease burden in animal models of atherosclerosis, thrombosis, heart failure, and chronic kidney disease (*Roberts et al., 2018*; *Wang et al., 2015*; *Gupta et al., 2020*; *Organ et al., 2020*). Although the gut microbial TMAO pathway has been studied mostly in the context of CVD (*Wang et al., 2011*; *Koeth et al., 2013*; *Zhu et al., 2016*; *Zhu et al., 2017*; *Tang et al., 2013*; *Wang et al., 2014b*; *Trøseid et al., 2015*; *Tang and Hazen, 2014*), recent studies found that breath levels of the primary metabolite TMA and other related co-metabolites are elevated in patients with ALD (*Hanouneh et al., 2014*; *Ascha et al., 2016*). These data showed promise, but whether the gut microbial TMAO pathway is causally related to ALD has never been explored. Hence, here we set out to understand how the gut microbial TMA/TMAO pathway may play a contributory role in ALD susceptibility and progression, and to test whether selective drugs that lower gut microbial production of TMA can be an effective therapeutic strategy. In an era when host genetics/genomics approaches dominate, this work reminds us that genes and metabolic products produced by gut bacteria play equally important roles in modulating disease susceptibility. Whereas pathways encoded by the host genome have long been pursued as drug targets, this work provides proof of concept that rationally designed drugs that target bacterial metabolism likely have untapped therapeutic potential in ALD and beyond.

## Results

### Circulating levels of the gut microbial metabolite TMA are elevated in AH

In a previous collaborative study, we reported that the highly volatile microbial metabolite TMA is elevated in exhaled breath of patients with AH (*Hanouneh et al., 2014*), and related co-metabolites, such as trimethyllysine and carnitine, can serve as prognostic indicators of mortality in AH (*Ascha et al., 2016*). Given the extremely volatile nature of TMA, it is readily detectable in breath, but is challenging to accurately quantitate levels in the circulation because TMA rapidly dissipates during collection and storage. To reduce the volatility of TMA and enable its analysis in the circulation, we coordinated patient blood collection utilizing rapid acidification of separated plasma (protonated TMA has a lower vapor pressure) across a large multi-center AH consortium (Defeat Alcoholic Steatohepatitis [DASH] consortium) (*Crabb et al., 2016*; *Vatsalya et al., 2020*; *Saha et al., 2019*). This provided us the unique opportunity to accurately quantify circulating TMA levels in human subjects, including those with moderate or severe AH for the first time. Patient demographics and clinical characteristics for the cohort examined are summarized in *Figure 1—source data 1*; *Figure 1—source data 2*. Importantly, MELD score, Maddrey's discriminant function score, Child-Pugh score, aspartate aminotransferase (AST), total bilirubin, creatinine, and international normalized ratio were higher in patients with severe AH compared to moderate AH patients, while serum albumin was lower in severe AH compared to

moderate AH patients. In agreement with previous breath metabolomics studies (*Hanouneh et al., 2014*; *Ascha et al., 2016*), plasma TMA levels were significantly elevated in moderate and severe AH patients compared to healthy controls (*Figure 1A*). However, the CVD-related co-metabolite TMAO was reciprocally decreased in AH patients (*Figure 1B*). Given the reciprocal alterations in plasma TMA and TMAO levels, we next examined the expression of the host liver enzyme flavin-containing mono-oxygenase 3 (FMO3) which is the predominant TMA to TMAO converting enzyme in the adult liver (*Cashman, 2002*). Interestingly, mRNA levels for FMO3 are uniquely repressed in patients with more severe AH (AH with liver failure [MELD 22–28] and AH with emergency liver transplant [MELD 18–21]), but not in other liver disease etiologies such as non-alcoholic fatty liver disease (NAFLD) or viral hepatitis (*Figure 1C*). In agreement with reduced mRNA levels (*Figure 1C*), patients with severe AH undergoing emergency liver transplant have marked reduction in FMO3 protein (*Figure 1D*), which likely contributes to elevations in plasma TMA (*Figure 1A*). Although ethanol feeding in mice does not consistently result in reduced hepatic *Fmo3* expression (data not shown), a single injection of lipo-polysaccharide (LPS) to induce acute hepatic inflammation is associated with both a reduction in the expression of *Fmo3* and a significant increase in the TMA receptor trace amine-associated receptor 5 (*Taar5*) (*Figure 1E*). It is important to note that circulating choline levels was not significantly altered in patients with AH compared to healthy controls (*Figure 1—figure supplement 1*). However, plasma levels of one of the gut microbial substrates for TMA production (carnitine) and other TMA pathway co-metabolites (e.g. betaine and γ-butyrobetaine) were elevated in patients with AH compared to healthy controls (*Figure 1—figure supplement 1*). These findings, in addition to previous breath metabolomic studies (*Hanouneh et al., 2014*; *Ascha et al., 2016*), provide evidence that TMA and related co-metabolites may allow for discrimination of AH from other liver diseases.

## Microbial choline TMA lyase inhibition protects mice from ethanol-induced liver injury

We next sought to establish whether a causal relationship between gut microbial TMA production and ALD progression exists, and to test the hypothesis that selectively drugging microbial choline trans-formation can serve as a mechanism for improving host liver disease and attenuating ethanol-induced liver injury in mice. Mice were individually treated with two recently reported non-lethal bacterial choline TMA lyase inhibitors, iodomethylcholine (IMC) and fluoromethylcholine (FMC) (*Roberts et al., 2018*). These small molecule inhibitors exhibit potent in vivo inhibition of the gut microbial choline TMA lyase enzyme CutC (*Craciun and Balskus, 2012*), and have been shown to effectively block bacterial choline to TMA conversion in vivo (*Roberts et al., 2018*). Designed as suicide substrate mechanism-based inhibitors, past studies reveal that the vast majority of IMC and FMC is retained in the gut within luminal bacteria and excreted in the feces with limited systemic exposure of the polar drug in the host (*Roberts et al., 2018*; *Gupta et al., 2020*; *Organ et al., 2020*).

IMC treatment effectively blunted ethanol-induced increases in plasma TMA and TMAO (*Figure 2A and B*). IMC also produced modest increases in plasma choline and betaine, while reducing plasma carnitine, particularly in pair-fed mice (*Figure 2C–E*). IMC also prevented ethanol-induced increases in alanine aminotransferase (ALT) and hepatic steatosis (*Figure 2F, G, and K*). Interestingly, IMC treatment prevented ethanol-induced increases in hepatic triglycerides (*Figure 2G*), and reduced hepatic total and cholesterol esters, but not free cholesterol, in both pair- and ethanol-fed conditions (*Figure 2H–I*). IMC treatment also reduced the expression levels of the pro-inflammatory cytokine tumor necrosis factor α (*Tnfα*) (*Figure 2L*). Although IMC was well tolerated in several previous mouse studies in the setting of standard rodent chow-feeding (*Roberts et al., 2018*; *Gupta et al., 2020*; *Organ et al., 2020*), here we found an unexpected reduction in food intake and body weights in mice receiving both IMC and ethanol (*Figure 2—figure supplement 1A, B*). Although IMC was clearly protective against ethanol-induced liver injury, this potential drug-ethanol interaction prompted us to test another structurally distinct gut microbe-targeted choline TMA lyase inhibitor FMC (*Roberts et al., 2018*; *Figure 3* and *Figure 3—figure supplement 1*).

Importantly, FMC was well tolerated and did not significantly alter liquid diet intake or body weights throughout the 25-day chronic ethanol feeding study (*Figure 2—figure supplement 1C, D*). FMC treatment trended toward reducing plasma TMA (*Figure 3A*), and more dramatically suppressed plasma TMAO levels (*Figure 3B*). Unlike IMC, which also altered other co-metabolites such as choline, betaine, and carnitine (*Figure 2B-E*), FMC did not significantly alter these TMA co-metabolites

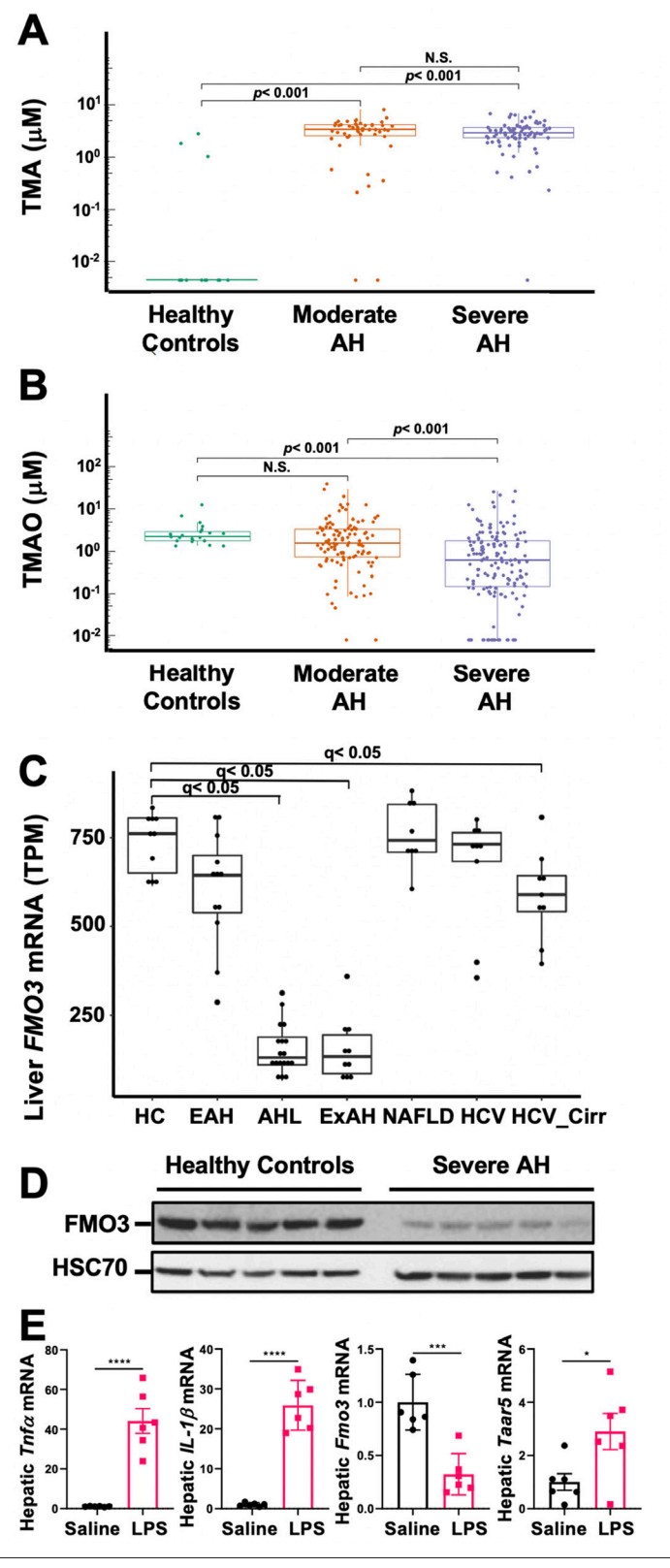

**Figure 1.** The gut microbial volatile metabolite trimethylamine (TMA) is elevated in alcohol-associated hepatitis (AH). Plasma TMA (**A**) and trimethylamine N-oxide (TMAO) (**B**) levels in patients considered healthy (n = 13 for TMA and 20 for TMAO), or who have moderate (MELD < 20) (n = 52 for TMA and 111 for TMAO) or severe (MELD > 20) (n = 83 for TMA and 152 for TMAO) AH. (**C**) RNA sequencing results from liver tissues of patients with different

*Figure 1 continued on next page*

*Figure 1 continued*

pathologies, including: healthy controls (HC, n = 10), early AH (EAH, n = 12; MELD 7–8), AH with liver failure (AHL, n = 18; MELD 22–28), explant tissue from patients with severe AH with emergency liver transplants (ExAH, n = 10; MELD 18–21), non-alcohol-associated fatty liver disease (NAFLD; n = 8), hepatitis C virus (HCV; n = 9), and hepatitis C virus with cirrhosis (HCV_Cirr, n = 9). Gene expression was measured by transcripts per million (TPM). Boxplots of average expression for *Fmo3* in different disease groups; error bars indicate SD (q < 0.05 in comparison to healthy controls). (**D**) Liver FMO3 protein expression measured by Western blot from healthy patients and patients with severe AH undergoing emergency liver transplant (Maddrey's discriminant function 45–187). (**E**) Liver *Tnfa*, *Il1b*, *Fmo3,* and *Taar5* transcript levels were measured by qPCR from female WT mice injected with either saline or lipopolysaccharide (LPS) for 6 hr. N = 6; unpaired Student's t-test. *p ≤ 0.05; ***p ≤ 0.001.

The online version of this article includes the following source data and figure supplement(s) for figure 1:

**Source data 1.** Demographic and clinical parameters for entire cohort of healthy controls and patients with AH.

**Source data 2.** Demographic and clinical parameters for subset of healthy controls and patients with AH included in TMA assay.

**Source data 3.** Liver flavin-containing monooxygenase 3 (FMO3) protein expression measured by Western blot from healthy patients (HC) and patients with severe alcohol-associated hepatitis (AH) undergoing emergency liver transplant (Maddrey's discriminant function 45–187).

**Source data 4.** Liver flavin-containing monooxygenase 3 (FMO3) protein expression measured by Western blot from healthy patients (HC) and patients with severe alcohol-associated hepatitis (AH) undergoing emergency liver transplant (Maddrey's discriminant function 45–187).

**Source data 5.** Liver HSC70 protein expression measured by Western blot from healthy patients (HC) and patients with severe alcohol-associated hepatitis (AH) undergoing emergency liver transplant (Maddrey's discriminant function 45–187).

**Source data 6.** Liver HSC70 protein expression measured by Western Blot from healthy patients (HC) and patients with severe alcohol-associated hepatitis (AH) undergoing emergency liver transplant (Maddrey's discriminant function 45–187).

**Figure supplement 1.** Levels of trimethylamine (TMA)-related metabolites in alcohol-associated hepatitis (AH).

(*Figure 3C-E*). More importantly, as with IMC (*Figure 2F–K*), FMC treatment significantly protected against ethanol-induced ALT elevations (*Figure 3F*), hepatic steatosis (*Figure 3G and K*), and reduced total and esterified cholesterol levels without altering free cholesterol (*Figure 3H–J*). However, FMC trended to reduce but did not significantly alter *Tnfα* expression (*Figure 3L*). To determine whether these effects were generalizable in other models of ethanol-induced liver injury, we exposed control and FMC-treated mice to a 10-day chronic model in which mice were allowed free access to a 5% vol/vol (27% kcal) for 10 days (*Figure 3—figure supplement 1*). In this 5%–10-day ethanol feeding model FMC treatment did not significantly alter food intake, body weight, or blood ethanol levels, but was able to selectively suppress TMA and TMAO levels (*Figure 3—figure supplement 1*). FMC treatment in the 5%–10-day model significantly reduced plasma AST and ALT levels, and trended toward lowering liver triglycerides (*Figure 3—figure supplement 1*). However, in this short-term model there were no apparent differences in hepatic cytokine/chemokine gene expression with either ethanol exposure or FMC treatment (*Figure 3—figure supplement 1* and data not shown). Collectively, these data demonstrate that gut microbe-targeted choline TMA lyase inhibition with two structurally distinct inhibitors (IMC or FMC) can generally protect mice against ethanol-induced liver injury.

## Microbial choline TMA lyase inhibitors promote remodeling of the gut microbiome and host liver transcriptome in an ethanol-dependent manner

One theoretical advantage of the selective microbe-targeted choline TMA lyase inhibitors, compared to antibiotic or MAMP-PRR-targeted therapies, is that they are anticipated to exert less selective pressure for development of drug resistance given their non-lethal nature. However, microbes that preferentially utilize choline as a carbon or nitrogen source might be anticipated to have reduced competitive advantage in the presence of the inhibitor. We therefore next examined whether IMC or FMC treatment was associated with alterations in choline utilizers and other members of the murine gut microbiome community that are known to be correlated with ethanol-induced liver injury (*Chen*

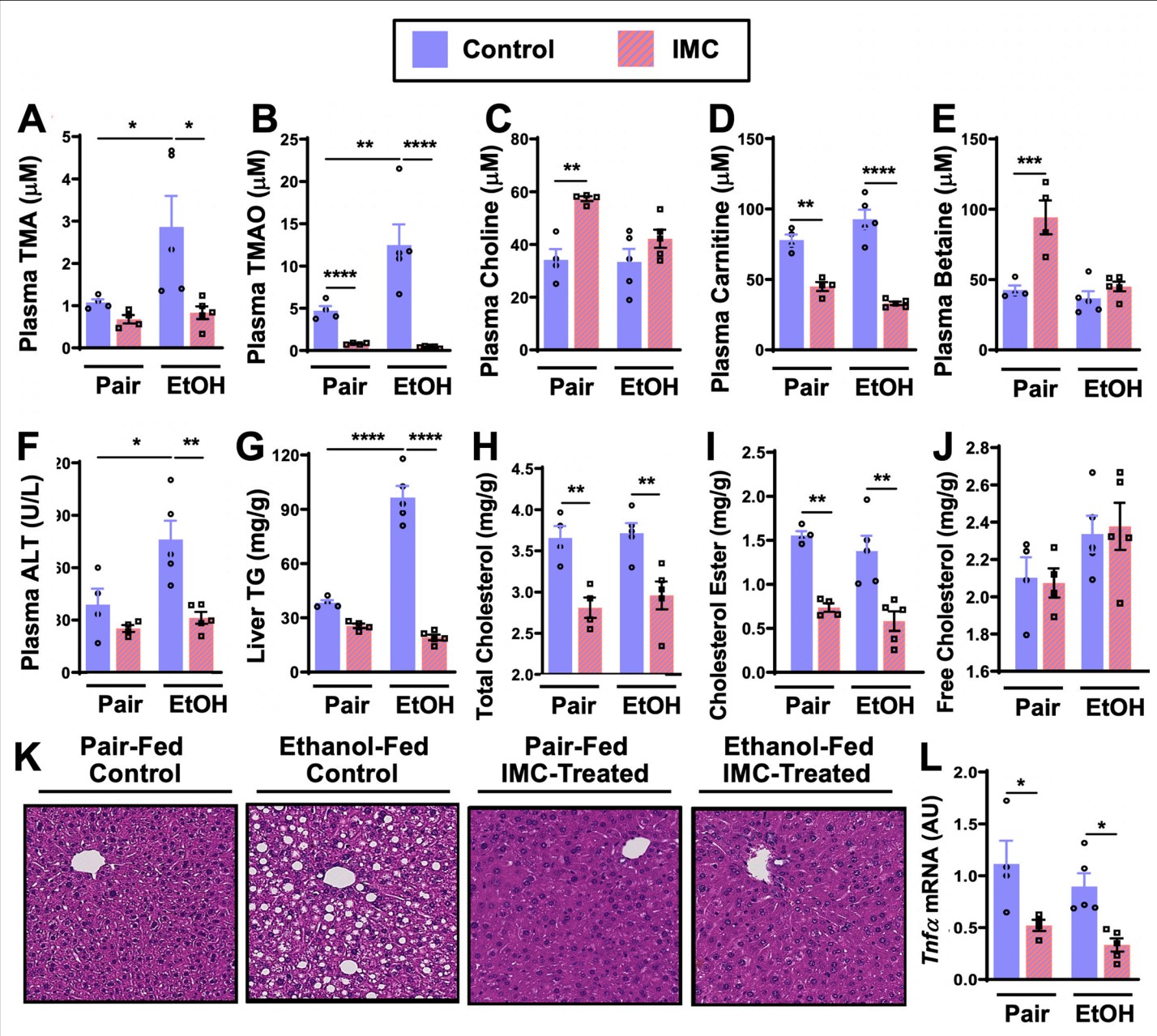

**Figure 2.** Small molecule choline trimethylamine (TMA) lyase inhibition with iodomethylcholine (IMC) protects mice against ethanol-induced liver injury. Nine- to eleven-week-old female C57BL6/J mice were fed either ethanol-fed or pair-fed in the presence and absence of IMC as described in the methods. Plasma levels of TMA (**A**), trimethylamine N-oxide (TMAO) (**B**), choline (**C**), carnitine (**D**), and betaine (**E**) were measured by mass spectrometry (n = 4–5). Plasma alanine aminotransferase (ALT) (**F**) was measured enzymatically (n = 4–5). Liver triglycerides (**G**), total cholesterol (**H**), cholesterol esters (**I**), and free cholesterol (**J**) were measured enzymatically (n = 4–5). (**K**) Representative H&E staining of livers from pair and EtOH-fed mice in the presence and absence of IMC. (**L**) Hepatic messenger RNA levels of tumor necrosis factor alpha (*Tnfα*). Statistics were completed by a two-way analysis of variance (ANOVA) followed by a Tukey's multiple comparison test. *p ≤ 0.05; **p ≤ 0.01; ***p ≤ 0.001; ****p ≤ 0.0001. All data are presented as mean ± SEM, unless otherwise noted.

The online version of this article includes the following figure supplement(s) for figure 2:

**Figure supplement 1.** Small molecule inhibition with iodomethylcholine (IMC), but not fluoromethylcholine (FMC), reduces food intake in ethanol-fed mice.

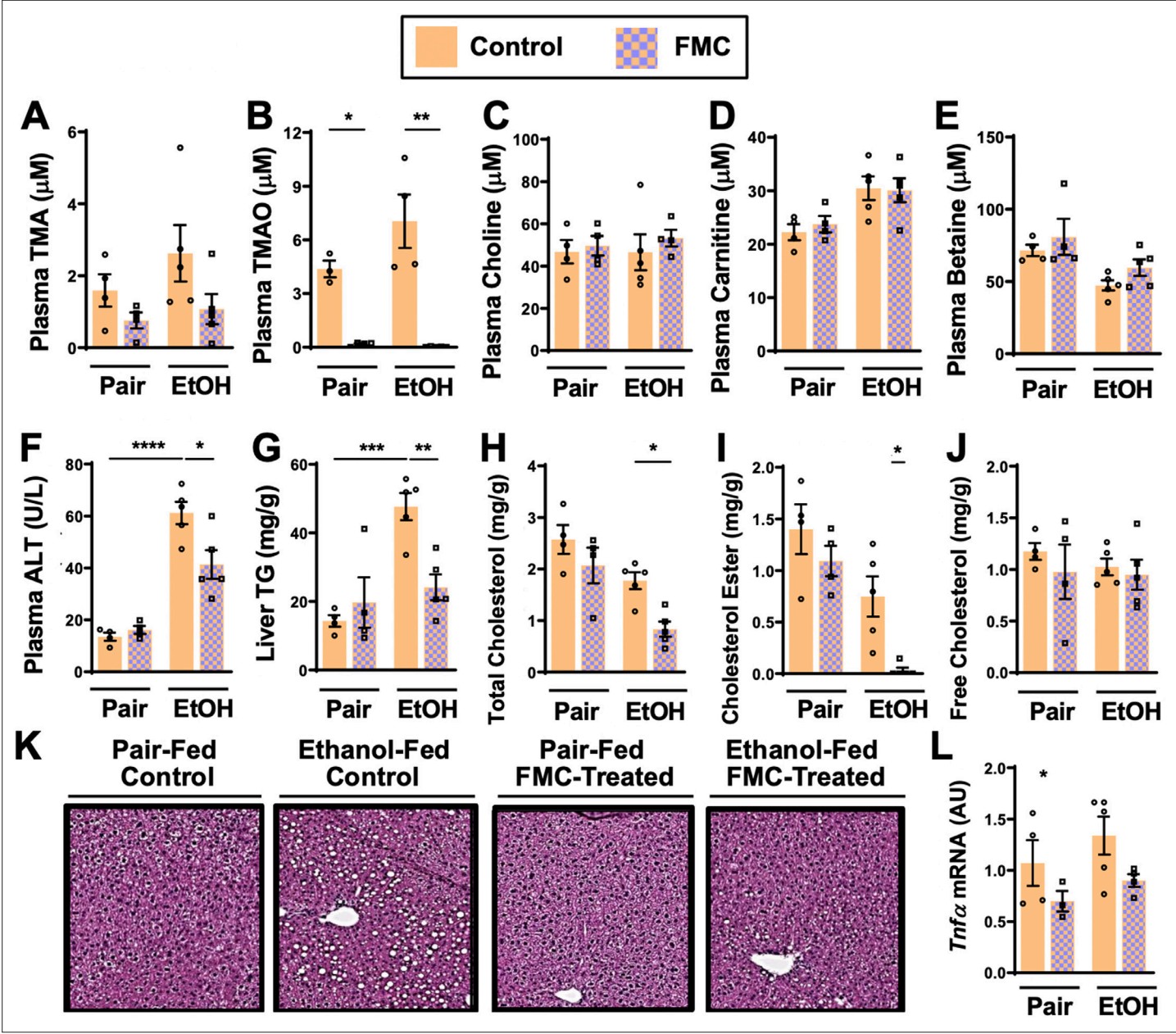

**Figure 3.** Small molecule choline trimethylamine (TMA) lyase inhibition with fluoromethylcholine (FMC) protects mice against ethanol-induced liver injury. Nine- to eleven-week-old female C57BL6/J mice were fed either ethanol-fed or pair-fed in the presence and absence of FMC as described in the methods. Plasma levels of TMA (**A**), trimethylamine N-oxide (TMAO) (**B**), choline (**C**), carnitine (**D**), and betaine (**E**) were measured by mass spectrometry (n = 3–5). Plasma alanine aminotransferase (ALT) (**F**) were measured at necropsy (n = 4–5). Liver triglycerides (**G**), total cholesterol (**H**), cholesterol esters (**I**), and free cholesterol (**J**) were measured enzymatically (n = 4–5). (**K**) Representative H&E staining of livers from pair and EtOH-fed mice in the presence and absence of FMC. (**L**) Hepatic messenger RNA levels of tumor necrosis factor alpha (*Tnfα*). Statistics were completed by a two-way analysis of variance (ANOVA) followed by a Tukey's multiple comparison test. *p ≤ 0.05; **p ≤ 0.01; ***p ≤ 0.001; ****p ≤ 0.0001. All data are presented as mean ± SEM, unless otherwise noted.

The online version of this article includes the following figure supplement(s) for figure 3:

**Figure supplement 1.** Small molecule inhibition of gut microbial trimethylamine (TMA) lyase activity with fluoromethylcholine (FMC) in a second model of ethanol-induced liver injury.

**Figure supplement 2.** A single bolus of ethanol does not significantly alter trimethylamine (TMA) or trimethylamine N-oxide (TMAO) levels in mice.

*et al., 2011*; *Yan et al., 2011*; *Mutlu et al., 2009*; *Mutlu et al., 2012*; *Tripathi et al., 2018*; *Ciocan et al., 2018*; *Llopis et al., 2016*; *Duan et al., 2019*; *Smirnova et al., 2020*; *Gao et al., 2019*; *Puri et al., 2018*; *Lang and Schnabl, 2020*). It is important to note that both IMC (*Figure 4A–E*) and FMC (*Figure 4F–J*) altered the gut microbiome, with some consistent, yet several distinct differences. Non-metric multidimensional scaling (NMDS) of microbial taxa revealed distinct clusters, indicating that both IMC and FMC promoted clear restructuring of the cecal microbiome in an ethanol-dependent manner (*Figure 4A and F*). Under pair-feeding conditions, both IMC and FMC caused a reciprocal decrease in the relative abundance of Bacteroidetes and increase in Firmicutes (*Figure 4B and G*). However, under ethanol-fed conditions IMC resulted in increased Bacteroidetes and reduced Firmicutes, and FMC treatment resulted in more modest reductions in Bacteroidetes and increased Firmicutes (*Figure 4B and G*). When examining drug-specific alterations at the genus level, we found that under both pair- and ethanol-fed conditions, IMC treatment promoted significant increases in *Faecalibaculum* and *Escherichia/Shigella*, and reductions in Bacteroidales_S24-7 (*Figure 4C–E , and H–I*). FMC, however, most significantly altered *Turicibacter*, *Oscillibacter*, and Lachnospiraceae, and it is important to note that these FMC-induced alterations were different between pair- and ethanol-fed groups (*Figure 4C–E , and H–I*). Collectively, these data demonstrate that inhibition of gut microbial choline to TMA transformation with a selective non-lethal small molecule inhibitor promotes restructuring of the gut microbiome in an ethanol-dependent manner.

To more globally understand the effects of choline TMA lyase inhibitors on the host liver, we performed unbiased RNA sequencing in mice undergoing pair or ethanol feeding treated either with or without IMC (*Figure 5*). NMDS and hierarchical clustering analysis showed clear separation between all four groups (*Figure 5A and B*). In pair-fed mice, IMC treatment caused significant decreases in several genes encoding major urinary proteins (*Mup2*, *Mup10*, *Mup11*, and *Mup18*) and cytochrome p450 enzymes (*Cyp3a16*, *Cyp3a44*), while increasing other genes involved in xenobiotic metabolism (*Ephx1*, *Cyp4a31*) and hormone/cytokine signaling (*Lepr*, *Fgf21*, *Il22ra1*) (*Figure 5C*). Under ethanol-feeding conditions, IMC treatment most significantly altered genes involved in hepatocyte metabolism (*Cyp8b1*, *Ugt1a5*, *Pnpla5*, *Sult2a8, Ces3a*, and *Cmah*), RNA processing (*Ddx21*, *Ftsj3*, *Dus1l, and Cmah*), and again major urinary proteins (*Mup2*, *Mup10*, *Mup11*, and *Mup20*) (*Figure 5D and E*). These unbiased RNASeq data demonstrate that gut microbe-targeted choline TMA lyase inhibitors can alter the host liver transcriptome in an ethanol feeding-dependent manner.

## The microbe-derived metabolite TMA elicits rapid hormone-like signaling effects in mouse liver

The gut microbe-derived co-metabolites TMA and TMAO are generated postprandially in both rodents and humans after a substrate-rich meal is ingested (*Schugar et al., 2018*; *Boutagy et al., 2015*). Given the acute meal-related production and recent identification of candidate host receptors for TMA (*Li et al., 2013*; *Wallrabenstein et al., 2013*) and TMAO (*Chen et al., 2019*), we hypothesized that TMA may be acting as a gut microbe-derived hormone to promote liver injury. However, currently nothing is known regarding the acute hormone-like signaling effects stimulated by TMA in the liver. To address this gap, we infused TMA directly into the portal circulation draining the gut (i.e. portal vein) of fasted mice and examined global phosphorylation events stimulated in the liver 10 min later using a phosphoproteomics approach (*Figure 6A*). It is important to note that this experiment provided high levels of exogenous TMA via direct injection, and future studies should focus on more physiologically relevant modes of TMA production like provision of gut bacteria that can naturally or be genetically engineered to produce high levels. A total of 36 liver proteins exhibited site-specific hypo- or hyper-phosphorylation 10 min after administration of TMA relative to vehicle-injected mice (*Figure 6B*). Several of the TMA-driven phosphorylation events represented proteins that are enriched in key hormonal signaling pathways known to impact hepatic metabolism. For example, portal vein infusion of TMA resulted in altered phosphorylation of proteins implicated in protein kinase A (PKA) signaling, including A kinase anchor protein 1 (AKAP1) (*Huang et al., 1999*) and FK506-binding protein 15 (FKBP15) (*Nooh and Bahouth, 2017*), and insulin signaling including insulin receptor substrate 2 (IRS2) (*Araki et al., 1994*; *Figure 6B–D*). TMA infusion was also associated with altering the phosphorylation of several guanine nucleotide exchange factors (GEF), including Rac/Cdc42 guanine nucleotide exchange factor 6 (Arhgef6) and Rho GTPase activating protein 17 (ARHGAP17) (*Zhou et al., 2016*; *Aslan, 2019*), and proteins involved in RNA processing/splicing including signal recognition

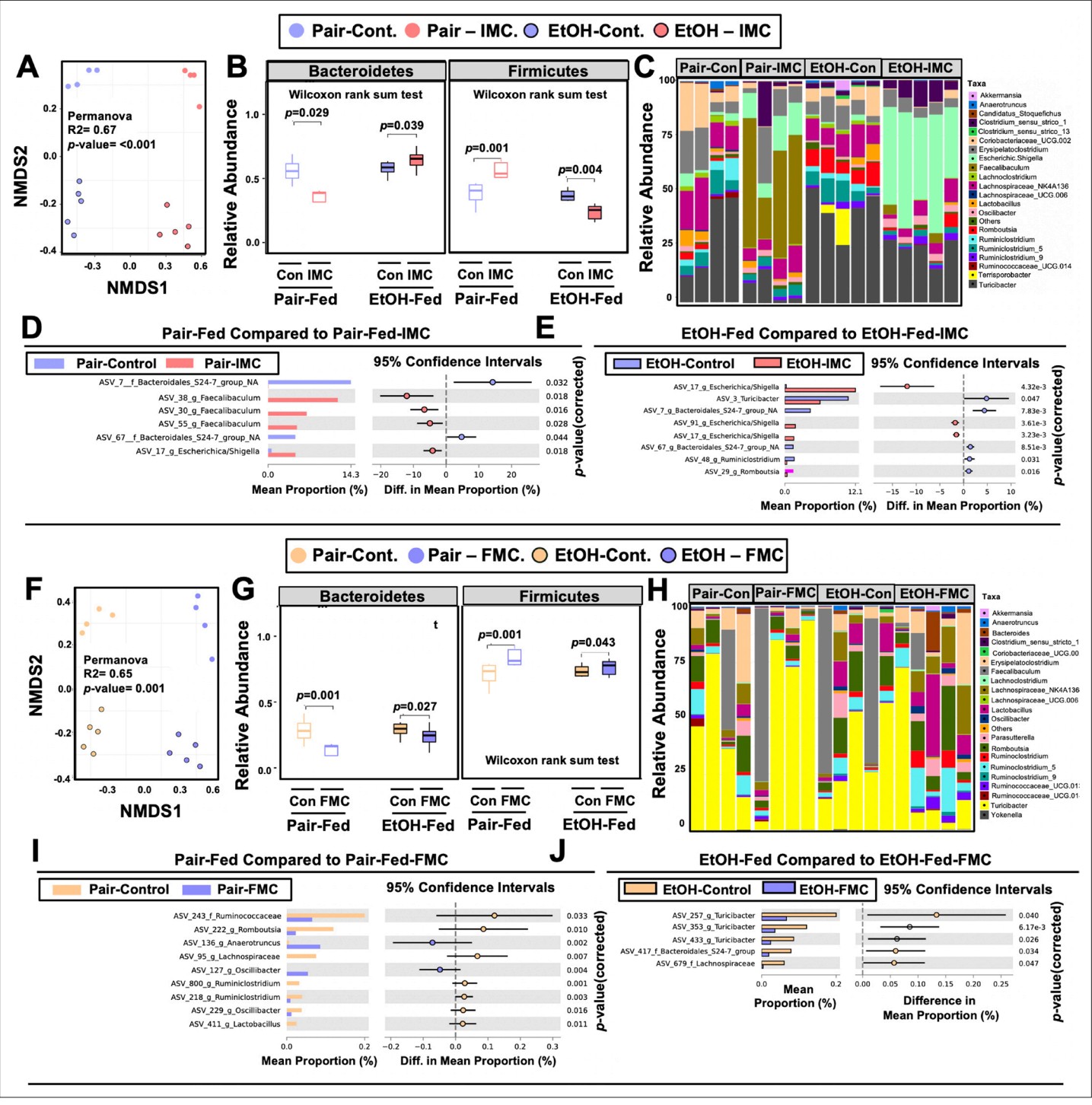

**Figure 4.** Small molecule choline trimethylamine (TMA) lyase inhibition promotes remodeling of the gut microbiome in an ethanol-dependent manner. Nine- to eleven-week-old female C57BL6/J mice were fed either ethanol-fed or pair-fed in the presence and absence of iodomethylcholine (IMC) or fluoromethylcholine (FMC) as described in the methods. (**A**) Non-metric multidimensional scaling (NMDS) plots based on the Bray-Curtis index between the pair, EtOH, pair + 0.06% IMC, and EtOH + 0.06% IMC groups, Statistical analysis was performed with permutational multivariate analysis of variance (PERMANOVA), and p-values are labeled in plots. $R^2$ values are noted for comparisons with significant p-values and stand for percentage variance explained by the variable of interest. (**B**) Boxplots of relative abundance patterns for Firmicutes and Bacteroidetes distinguishing pair, EtOH, pair + 0.06% IMC and EtOH + 0.06% IMC groups. Statistical analysis was performed with Mann-Whitney U test (also called the *Wilcoxon rank-sum test*, p-values are labeled in plots). Plotted are interquartile ranges (boxes), and dark lines in boxes are medians. (**C**) Stacked bar charts of relative abundance (left y-axis) of the top 20 genera assembled across all four groups (pair, EtOH, pair + 0.06% IMC, and EtOH + 0.06% IMC groups). Pairwise

*Figure 4 continued on next page*

*Figure 4 continued*

differential abundance analyses between (**D**) pair-fed and pair-fed + 0.06% IMC and (**E**) EtOH-fed and EtOH-fed + 0.06% IMC group. Statistical analysis was performed with White's non-parametric t-test (p-values are labeled in plots). (**F**) NMDS plots based on the Bray-Curtis index between the pair, EtOH, pair + 0.006% FMC, and EtOH + 0.006% FMC groups, Statistical analysis was performed with permutational multivariate analysis of variance (PERMANOVA), and p-values are labeled in plots. $R^2$ values are noted for comparisons with significant p-values and stand for percentage variance explained by the variable of interest. (**G**) Boxplots of relative abundance patterns for Firmicutes and Bacteroidetes distinguishing pair, EtOH, pair + 0.006% FMC, and EtOH + 0.006% FMC groups. Statistical analysis was performed with Mann-Whitney U test (also called the *Wilcoxon rank-sum test*, p-values are labeled in plots). Plotted are interquartile ranges (boxes), and dark lines in boxes are medians. (**H**) Stacked bar charts of relative abundance (left y-axis) of the top 20 genera assembled across all four groups (pair, EtOH, pair + 0.06% FMC, and EtOH + 0.006% FMC groups). Pairwise differential abundance analyses between (**I**) pair-fed and pair-fed + 0.06% FMC, and (**J**) EtOH-fed and EtOH-fed + 0.006% FMC group. Statistical analysis was performed with White's non-parametric t-test (p-values are labeled in plots).

particle 14 (SRP14) (*Strub and Walter, 1990*) and serine- and arginine-rich splicing factor 1 (SRSF1) (*Cho et al., 2011*; *Figure 6B and C*). These data have identified acute TMA-driven signaling events in the liver in vivo, and potentially link TMA to acute alterations in PKA-, insulin-, and GEF-driven signaling cascades that deserve further exploration.

## Discussion

Although drug discovery has historically targeted pathways in the human host, there is untapped potential in therapeutically targeting the gut microbial endocrine organ to treat advanced liver disease. This paradigm shift is needed in light of the clear and reproducible associations between the gut microbiome in viral, alcohol-associated, and non-alcohol-associated liver diseases (*Chen et al., 2011*; *Yan et al., 2011*; *Mutlu et al., 2009*; *Mutlu et al., 2012*; *Tripathi et al., 2018*; *Ciocan et al., 2018*; *Llopis et al., 2016*; *Duan et al., 2019*; *Smirnova et al., 2020*; *Gao et al., 2019*; *Puri et al., 2018*; *Lang and Schnabl, 2020*; *Wilkinson et al., 1974*; *Tarao et al., 1979*; *Uesugi et al., 2001*; *Paik et al., 2003*; *DeSantis et al., 2013*; *Knorr et al., 2020*; *Brown and Hazen, 2015*). Now we are faced with both the challenge and opportunity to test whether microbe-targeted therapeutic strategies can improve health in the human metaorganism without negatively impacting the symbiotic relationships between microbes and host. Although traditional microbiome manipulating approaches such as antibiotics, prebiotics, probiotics, and fecal microbial transplantation have shown their own unique strengths and weaknesses, each of these presents unique challenges particularly for use in chronic diseases such as end stage liver disease. As we move toward selective non-lethal small molecule therapeutics, the goal is to have exquisite target selectivity and limited systemic drug exposure given that the targets are microbial in nature. This natural progression parallels the paradigm shifts in oncology which have transitioned from broadly cytotoxic chemotherapies to target-selective small molecule and biologics-based therapeutics. Here, we provide the first evidence that the gut microbial choline metabolite TMA is elevated in the plasma of patients with AH, which corroborates previous reports showing that TMA is also prominent in the breath of patients with AH (*Hanouneh et al., 2014*). Hence, further studies are warranted to determine whether combined measures of breath and blood TMA can serve as a prognostic biomarker to accurately predict AH-related mortality. Here, we also show for the first time that a selective non-lethal small molecule drug that reduces bacterial production of TMA can prevent ethanol-induced liver injury in mice. We also demonstrate that direct administration of TMA can elicit rapid signaling effects in the liver, supporting the notion that gut microbial metabolites produced postprandially can act in an endocrine-like manner to alter host signal transduction and associated disease pathogenesis. Collectively, these studies suggest that selective drugs targeting the gut microbial TMA pathway may hold promise for treating AH.

As drug discovery advances in the area of small molecule non-lethal bacterial enzyme inhibitors, it is key to understand how these drugs impact microbial ecology in the gut and other microenvironments. As we have previously reported (*Roberts et al., 2018*; *Gupta et al., 2020*; *Organ et al., 2020*), gut microbe-targeted choline TMA lyase inhibitors (IMC and FMC) induced a significant remodeling of the cecal microbiome in mice. In the current studies there were some consistent, but many different cecal microbiome alterations when comparing IMC and FMC (*Figure 4*), yet both drugs similarly improved ethanol-induced liver injury. As small molecule bacterial enzymes inhibitors are developed, it will be extremely important to understand their effects on microbial ecology, and it is expected that some of the beneficial effects of these drugs will indeed originate from the restructuring of gut microbiome

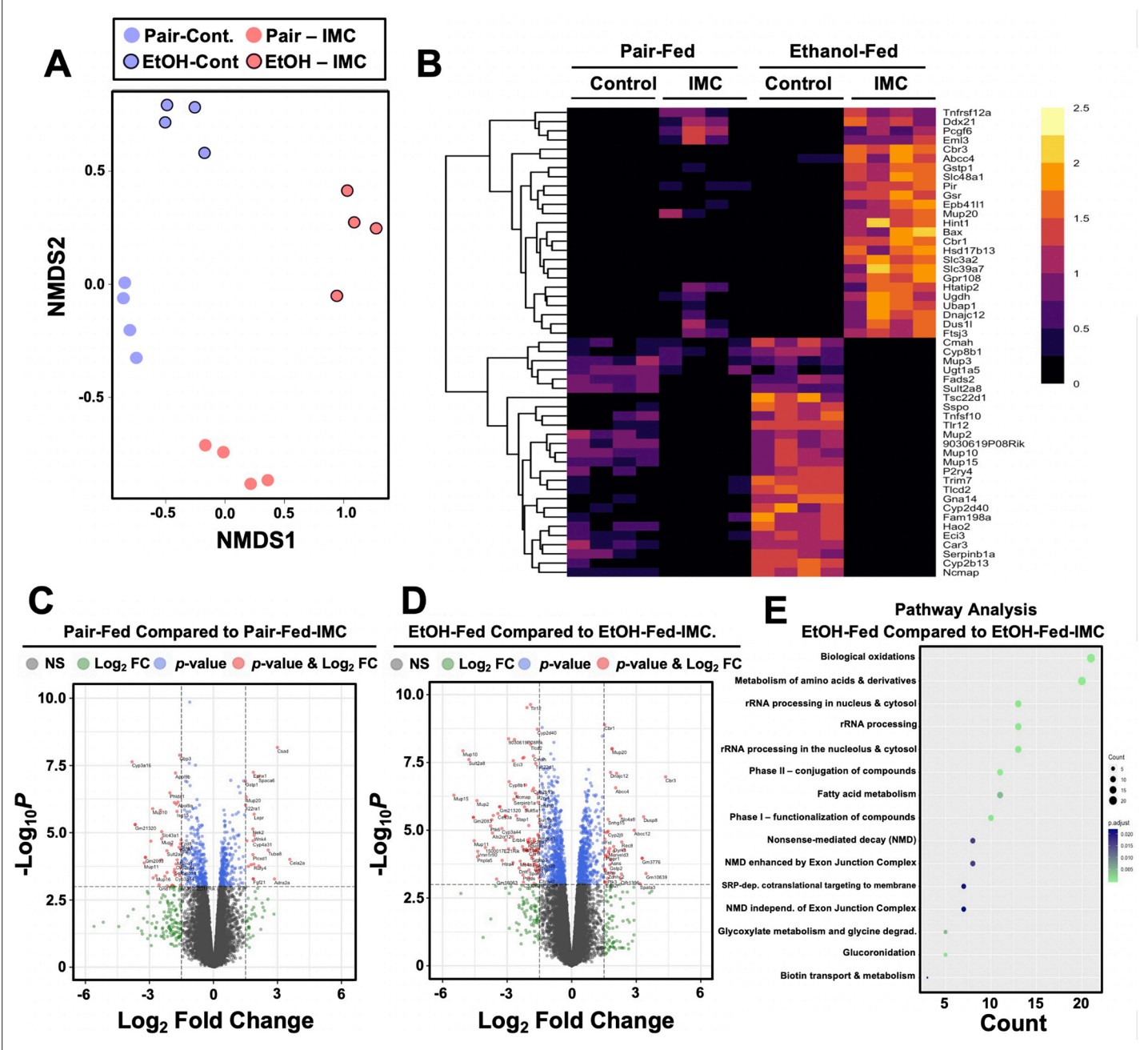

**Figure 5.** Small molecule choline trimethylamine (TMA) lyase inhibition with iodomethylcholine (IMC) alters the hepatic transcriptome in response to ethanol. Nine- to eleven-week-old female C57BL6/J mice were fed either ethanol-fed or pair-fed in the presence and absence of IMC as described in the methods. RNA was isolated from the livers and subjected to next-generation sequencing. (**A**) Non-metric multidimensional scaling (NMDS) plots; each point represents a single sample from a single mouse. Positions of points in space display dissimilarities in the transcriptome, with points further from one another being more dissimilar. (**B–C**) Row-normalized expression for the top 25 DEGs shown by heat map (**B**) while the volcano plot (**C**) summarizes log2 fold changes vs. significance in response to IMC treatment in pair (left) and ethanol (right) feeding (n = 4). (**D**) Summary of significantly differentially regulated pathways in mice treated with IMC in the ethanol-fed mice (n = 4).

communities. In fact, this is not an uncommon mechanism by which host targeted drugs impact human health. A recent study showed that nearly a quarter of commonly used host-targeted drugs have microbiome-altering properties (*Maier et al., 2018*), and in the context of diabetes therapeutics it is important to note metformin's anti-diabetic effects are partially mediated by the drug's microbiome altering properties (*Wu et al., 2017*). Given the strong association between gut microbiome and liver disease (*Chen et al., 2011*; *Yan et al., 2011*; *Mutlu et al., 2009*; *Mutlu et al., 2012*; *Tripathi et al.,*

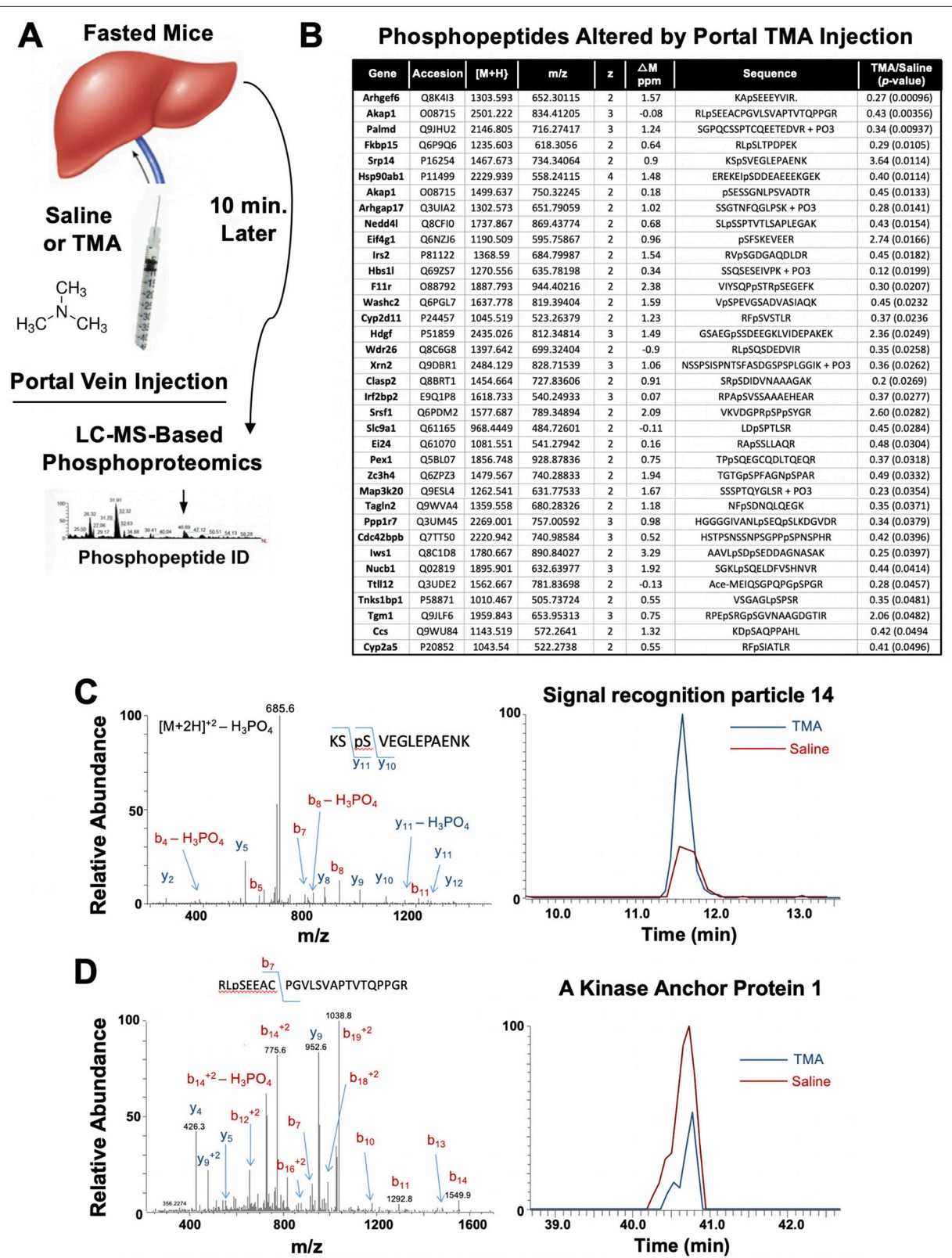

**Figure 6.** Trimethylamine (TMA) rapidly reorganizes liver signal transduction in vivo. (**A**) Schematic of experiment; female C57BL/6 mice were fasted overnight (12 hr fast), and then injected directly into the portal vein with vehicle (saline), or TMA, and only 10 min later liver tissue was harvested for phosphoproteomic analysis to identify TMA-responsive phosphorylation events in mouse liver (n = 4 per group). (**B**) List of proteins that were differentially phosphorylated (p < 0.05) upon TMA administration in vivo. (**C**) A doubly charged ion was present in the phospho-enriched sample

*Figure 6 continued on next page*

*Figure 6 continued*

that was identified as the KSpSVEGLEPAENK from signal recognition particle 14 kDa protein (Srp14). The CID spectra of this ion is dominated by $H_3PO_4$ loss from the precursor ion consistent with the presence of a pS or pT residue. The mass difference between the $y_{11}$ and $y_{10}$ ions is consistent with modification at S45. The observed chromatograms for this peptide from the saline and TMA samples are shown and the TMA/saline ratio was determined to be 3.6 (p-value 0.0114). (**D**) A doubly charged ion was present in the phospho-enriched sample that was identified as the RLpSEEACPGVLSVAPTVTQPPGR from A-kinase anchor protein 1. The CID spectra of this ion is dominated by fragmentation C-terminal to the proline residues. The mass of the $b_7$ ion is consistent with modification at S55. The observed chromatograms for this peptide from the saline and TMA samples are shown and the TMA/saline ratio was determined to be 0.4.

*2018*; *Ciocan et al., 2018*; *Llopis et al., 2016*; *Duan et al., 2019*; *Smirnova et al., 2020*; *Gao et al., 2019*; *Puri et al., 2018*; *Lang and Schnabl, 2020*; *Wilkinson et al., 1974*; *Tarao et al., 1979*; *Uesugi et al., 2001*; *Paik et al., 2003*; *DeSantis et al., 2013*; *Knorr et al., 2020*; *Brown and Hazen, 2015*; *Gilbert et al., 2018*), it may prove advantageous to find therapeutics that beneficially remodel the gut microbiome as well as engage either their microbe or host target of interest.

The metaorganismal TMA/TMAO pathway represents only one of many microbial metabolic circuits that have been associated with human disease (*Figure 7*). In fact, many microbe-associated metabolites such as short chain fatty acids, secondary bile acids, phenolic acids, polyamines, and others have more recently been associated with many human diseases (*Brown and Hazen, 2015*; *Gilbert et al., 2018*). In an ethanol- and meal-related manner, gut microbes produce a diverse array of metabolites that reach micromolar to millimolar concentrations in the blood, making the collective gut microbiome an active endocrine organ (*Brown and Hazen, 2015*). Small molecule metabolites are well known to be mediators of signaling interactions in the host, and this work provides evidence that diet/ethanol-microbe-host metabolic interplay can be causally linked to ethanol-induced liver injury. Our work, and that of many others, demonstrates that there is clear evidence of bi-directional crosstalk between the gut microbial endocrine organ and host liver metabolism. As drug discovery advances, it will be important to move beyond targets based solely in the human host. This work highlights that non-lethal gut microbe-targeted enzyme inhibitors may serve as effective therapeutics in AH and provides proof of concept that this may be a generalizable approach to target metaorganismal crosstalk in other disease contexts. In fact, selective inhibition of bacterial enzymes has the advantage over host targeting given that small molecules can be designed to avoid systemic absorption and exposure, thereby minimizing potential host off target effects. As shown here with the gut microbial TMA/TMAO pathway, it is easy to envision that other microbe-host interactions are mechanistically linked to host disease pathogenesis, serving as the basis for the rational design of microbe-targeted therapeutics that improve human health.

## Methods

**Key resources table**

| Reagent type (species) or resource | Designation | Source or reference | Identifiers | Additional information |
|---|---|---|---|---|
| Strain, strain background Mice (Females) | 9–11 Weeks | Jackson Laboratories | C57BL6/J, RRID:IMSR_ JAX:000664 | 5–8 per study |
| Biological sample (Humans) | Plasma samples from 285 patients | Cleveland Clinic Foundation; University of Louisville; University of Massachusetts Medical School; University of Texas Southwestern Medical Center | Not provided | |
| Biological sample (Humans) | Liver samples from five healthy donors | Clinical Resource for Alcoholic Hepatitis Investigations at Johns Hopkins University | Not provided | |
| Biological sample (Humans) | Liver samples from five patients with severe AH | Clinical Resource for Alcoholic Hepatitis Investigations at Johns Hopkins University | Not provided | |

*Continued on next page*

*Continued*

| Reagent type (species) or resource | Designation | Source or reference | Identifiers | Additional information |
|---|---|---|---|---|
| Antibody | Anti-FMO3 (Rabbit monoclonal) | Abcam | Cat# ab126790, RRID: AB_11128907 | 1:1000 (WB) |
| Antibody | Anti-HSC70 (Mouse monoclonal) | Santa Cruz Biotechnology | Cat# sc-7298, RRID: AB_627761 | 1:1000 (WB) |
| Antibody | Anti-rabbit IgG HRP | GE-Healthcare | Cat#: NA934-100UL, RRID: AB_772206 | 1:5000 (WB) |
| Antibody | Anti-mouse IgG HRP | GE-Healthcare | NA931V, RRID: AB_772210 | 1:5000 (WB) |
| Sequence-based reagent | Mouse Tnfα | Sigma | PCR primers | F:CCACCACGCTCTTCTGTCTAC R:AGGGTCTGGGCCATAGAACT |
| Sequence-based reagent | Mouse Il1β | Sigma | PCR primers | F:AGTTGACGGACCCCAAAAG R:AGCTGGATGCTCTCATCAGG |
| Sequence-based reagent | Mouse Fmo3 | Sigma | PCR primers | F:CCCACATGCTTTGAGAGGAG R:GGAAGAGTTGGTGAAGACCG |
| Sequence-based reagent | Mouse Taar5 | Sigma | PCR primers | F:AAAGAAAAGCTGCCAAGA R:AAGGGAAGCCAACACACA |
| Sequence-based reagent | Mouse CyclophilinA | Sigma | PCR primers | F:GCGGCAGGTCCATCTACG R:GCCATCCAGCCATTCAGTC |
| Sequence-based reagent | Mouse Cxcl1 | IDT | PCR primers | F:TGCACCCAAACCGAAGTC R:GTCAGAAGCCAGCGTTCACC |
| Sequence-based reagent | Mouse Grp78 | IDT | PCR primers | F:ACTTGGGGACCACCTATTCCT R:ATCGCCAATCAGACGCTCC |
| Commercial assay or kit | AST Commercial Kit | Sekisui Diagnostics | 319–30 | |
| Commercial assay or kit | ALT Commercial Kit | Sekisui Diagnostics | 318–30 | |
| Commercial assay or kit | Triglyceride Commercial Kit | Wako | 994–02891 | |
| Commercial assay or kit | Total Cholesterol Commercial Kit | Fisher Scientific | TR134321 | |
| Commercial assay or kit | Free Cholesterol Commercial Kit | Wako | 993–02501 | |
| Commercial assay or kit | RNAeasy Lipid Tissue Mini Kit | Qiagen | 74804 | |
| Commercial assay or kit | Thermo Scientific Pierce TiO$_2$ Phosphopeptide Enrichment and Clean-up Kit | Fisher Scientific | PI88301 | |
| Commercial assay or kit | RNAeasy Purification Kit | Qiagen | 74004 | |
| Chemical compound, drug | Iodomethylcholine (IMC) | Synthesized at the Cleveland Clinic | Not provided | |
| Chemical compound, drug | Fluoromethylcholine (FMC) | Synthesized at the Cleveland Clinic | Not provided | |
| Chemical compound, drug | Trimethylamine Hydrochloride | Sigma | T72761 | |
| Chemical compound, drug | Lipopolysaccharide | Sigma | L4391 | |

*Continued on next page*

*Continued*

| Reagent type (species) or resource | Designation | Source or reference | Identifiers | Additional information |
|---|---|---|---|---|
| Software, algorithm | GraphPad Prism | GraphPad Software, Inc | 8.4 | |
| Software, algorithm | DADA2 | https://benjjneb.github.io/dada2/dada-installation.html; *Callahan et al., 2016* | 1.16 | |
| Software, algorithm | Phyloseq | https://www.bioconductor.org/packages/release/bioc/html/phyloseq.html | 4.1, RRID:SCR_013080 | |
| Software, algorithm | microbiomeSeq | https://github.com/umerijaz/microbiomeSeq | 1: RRID:SCR_002630 | |
| Software, algorithm | Ggplot2 | https://cran.r-project.org/web/packages/ggplot2/index.html | 3.3.5, RRID:SCR_014601 | |
| Software, algorithm | vegan | https://cran.r-project.org/web/packages/vegan/index.html | 2.5–7 | |
| Other | Supersignal West Pico Plus Substrate | Thermo Fisher | 34577 | |
| Other | Diet | Dyets | 710260 | |

## Overview of human study populations

We made use of three different human study populations, detailed below, that included patients with severities of AH/ALD. It must be noted that one limitation of our study is that each of these cohorts used slightly different diagnostic criteria for defining the severity/stage of AH/ALD.

## Human study populations and sample collection for TMA measurement

A total of 285 subjects were included in this study. De-identified plasma samples, along with clinical and demographic data, were obtained from (1) the Northern Ohio Alcohol Center (NOAC) at the Cleveland Clinic biorepository including 21 healthy individuals and 15 patients diagnosed and (2) the Defeat Alcoholic Steatohepatitis (DASH) consortium (Cleveland Clinic, University of Louisville School of Medicine, University of Massachusetts Medical School, and University of Texas Southwestern Medical Center) including 249 patients with AH. Diagnosis with AH was performed using clinical and laboratory criteria, with MELD score utilized for distinguishing moderate (MELD < 20) and severe (MELD > 20) AH, as recommended by the NIAAA Alcoholic Hepatitis consortia (*Crabb et al., 2016*). A detailed description of patient recruitment, inclusion and exclusion criteria for the DASH consortium has been reported in previous studies (*Vatsalya et al., 2020*). Patients with AH were classified as moderate (MELD < 20, n = 112) and severe (MELD ≥ 20, n = 152) according to the MELD score at admission as part of either of two independent clinical trials (ClincalTrials.gov identifier # NCT01809132 and NCT03224949) or the NOAC biorepository. These studies were approved by the Institutional Review Boards of all four participating institutions and all study participants consented prior to collection of data and blood samples. Clinical and demographic data for the entire cohort is presented in *Figure 1—source data 1* and for the sub-set of subjects used for TMA analysis is presented in *Figure 1—source data 2*.

In order to be able to measure volatile compounds such as TMA, blood was collected in EDTA-coated tubes and immediately placed on ice. Plasma was separated by centrifugation at 1200× *g* for 15 min at 4°C. Plasma was rapidly acidified by adding 25 mL of 1 M hydrochloric acid (HCL) to 500 mL of aliquoted plasma, followed by vigorous vortexing. Acidified plasma samples were stored at –80°C in air-tight O-ring cryovials (Fisher Scientific, product # 02-681-373) until being processed for quantification of TMA and other volatile compounds. A non-acidified sample was also collected for standard plasma biochemistries.

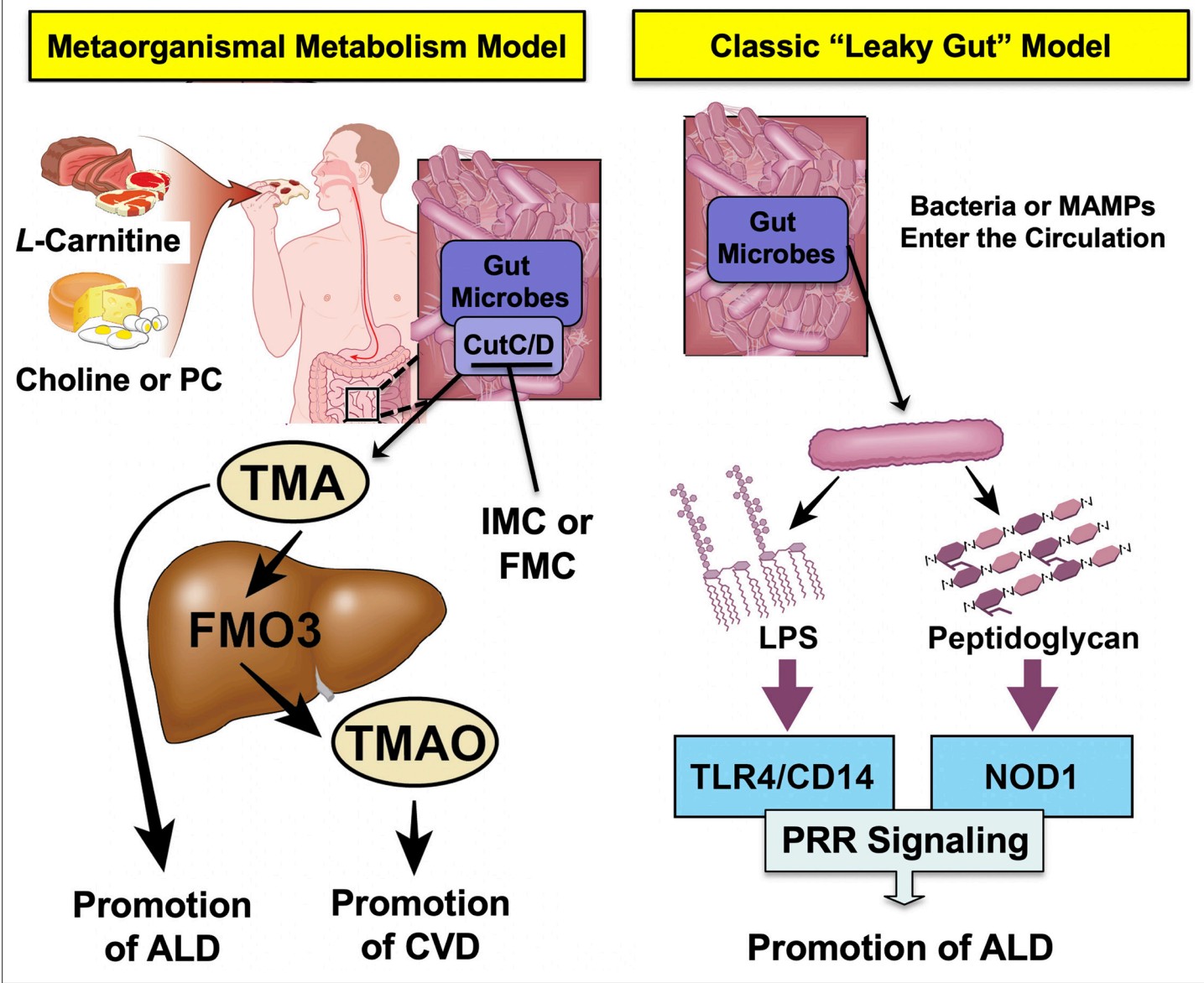

**Figure 7.** Graphical summary depicting the proposed role of trimethylamine (TMA) in the progression of alcohol-associated liver disease (ALD). Gut microbiota can elicit both metabolism-dependent and metabolism-independent effects in ALD. Relevant to this manuscript, intestinal microbes metabolize dietary L-carnitine, choline, or phosphatidylcholine (PC) to form TMA, which is a volatile compound that originates exclusively from gut bacterial metabolism and is elevated in ALD. Importantly, TMA can also be converted to trimethylamine N-oxide (TMAO) by hepatic flavin monooxygenase 3 (FMO3), and TMAO has recently been linked to cardiovascular disease (CVD) promotion in humans. Metabolism-independent effects are the result of gut hyperpermeability (leaky gut), allowing bacterial cell wall products such as lipopolysaccharide (LPS) and peptidoglycans to enter into the blood stream and engage with host pattern recognition receptors (PRR) to promote hepatic inflammation. Collectively, metabolism-dependent pathways such as TMA production as well as metabolism-independent pathways provide multiple bacterially derived 'hits' to promote ALD progression. The small molecule bacterially targeted CutC/D inhibitors iodomethylcholine (IMC) and fluoromethylcholine (FMC) can effectively blunt ethanol-induced liver injury in mice.

### Analysis of hepatic FMO3 expression across different liver disease etiologies

For data shown in main *Figure 1* panel C, we leveraged access to publicly available bulk liver RNA sequencing data from patients with different liver disease etiologies (*Argemi et al., 2019*). For this cohort, early AH (EAH) was defined as MELD 7–8, severe AH with liver failure (AHL) with MELD 22–28, and AHL with emergency liver transplant (ExAH) with MELD 18–21. All raw fastq files were downloaded from SRA (PRJNA531223) and dbGAP (phs001807.v1.p1) (*Argemi et al., 2019*). Fastq files

were aligned to the human genome (GRCh38, indices downloaded from https://github.com/pach-terlab/kallisto-transcriptome-indices/releases/download/ensembl-96/homo_sapiens.tar.gz; *Pachter, 2018*) using Kallisto version 0.44.0 with 100 bootstraps calculated (*Bray et al., 2016*). Data were then merged with clinical data and analyzed with Sleuth in gene_mode with aggregation_column set to Ensemble Gene ID; in addition, extra_bootstrap_summary and read_bootstrap_tpm were set to true (*Pimentel et al., 2017*). Differential expression was measured with Sleuth using a cutoff of q < 0.05.

## Human study populations and sample collection for liver Western blotting

De-identified samples from five livers explanted from severe AH patients during liver transplantation or five wedge biopsies from healthy donor livers were snap-frozen in liquid nitrogen and stored at –80°C. Samples were provided by the Clinical Resource for Alcoholic Hepatitis Investigations at Johns Hopkins University (R24 AA0025107, Z. Sun PI). Written informed consent was obtained from each patient included in the study and the study protocol conforms to the ethical guidelines of the 1975 Declaration of Helsinki as reflected in a priori approval by the Institutional Review Boards at Johns Hopkins Medical Institutions. This cohort utilized Maddrey's Discriminant Function as the primary indicator of disease severity, with an average score of 102.5 ± 27.7. MELD scores are not available for this cohort. Descriptive biochemical and clinical data for this cohort have been reported previously (*Tripathi et al., 2018*).

## Immunoblotting

Whole tissue homogenates were made from tissues in a modified RIPA buffer as previously described (*Warrier et al., 2015*; *Helsley et al., 2019*; *Schugar et al., 2017*; *Lord et al., 2016*), and protein was quantified using the bicinchoninic assay (Pierce). Proteins were separated by 4–12% SDS-PAGE, transferred to polyvinylidene difluoride membranes, and then proteins were detected after incubation with specific antibodies as previously described (*Warrier et al., 2015*; *Helsley et al., 2019*; *Schugar et al., 2017*; *Lord et al., 2016*) and listed in the Key resources table.

## Real-time PCR analysis of gene expression

Tissue RNA extraction and qPCR analysis was performed as previously described (*Helsley et al., 2019*). The mRNA expression levels were calculated based on the ΔΔ-CT method using cyclophilin A as the housekeeping gene. qPCR was conducted using the Applied Biosystems 7500 Real-Time PCR system. All primer sequences are listed in the Key resources table.

## Chemical synthesis of gut microbe-targeted choline TMA lyase inhibitors

The small molecule choline TMA lyase inhibitors IMC and FMC have been previously described as potent and selective mechanism-based inhibitors targeted microbial CutC (*Roberts et al., 2018*). Here, IMC and FMC were synthesized and structurally characterized as outlined below using both multinuclear NMR analysis and high-resolution mass spectrometry. $^{1}$H- and $^{13}$C-NMR spectra for IMC and FMC were recorded on a Bruker Ascend spectrometer operating at 400 MHz. Chemical shifts are reported as parts per million (ppm). IMC iodide was prepared using a previously reported method using 2-dimethylethanolamine and diiodomethane as reactants in acetonitrile followed by recrystallization from dry ethanol. $^{1}$H- and $^{13}$C-NMRs of IMC were both consistent with that in the reported literature (*Mistry et al., 2002*), as well as consistent based on proton and carbon chemical shift assignments indicated below. High-resolution MS corroborated the expected cation mass and provided further evidence of structural identity. $^{1}$H-NMR (400 MHz, $D_2O$): 5.24 (s, 2H, -N-C$\underline{H}_2$-I), 4.06–3.99 (m, 2H, -CH$_2$-C$\underline{H}_2$-OH), 3.68–3.62 (m, 2H, -N-C$\underline{H}_2$-CH$_2$-), 3.29 (s, 6H, -N(C$\underline{H}_3$)$_2$); $^{13}$C-NMR (100 MHz, $D_2O$): 66.1 (-CH$_2$-$\underline{C}$H$_2$-OH), 55.8 (-N-$\underline{C}$H$_2$-CH$_2$-), 52.9 (-N($\underline{C}$H$_3$)$_2$), 33.0 (-N-$\underline{C}$H$_2$-I); HRMS (ESI/TOF): m/z (M$^+$) calculated for $C_5H_{13}INO$, 230.0036; found, 230.0033. The synthesis of fluoromethylcholine chloride was performed using the procedure below. $^{1}$H- and $^{13}$C-NMRs of FMC were consistent with that in the reported literature (*Gao et al., 2019*). High-resolution MS was also consistent with the expected cation mass. Chloro(fluoro)methane (2.05 kg, 29.9 mol, 6 eq) was bubbled into a solution of 2-dimethylaminoethanol (444.0 g, 4.98 mol, 500 mL, 1 eq) in THF (1000 mL) at –70°C for 4 hr. The mixture was then transferred to an autoclave and heated to 80°C and stirred for 18 hr (pressure:

~15–50 psi). During this period, a white precipitate formed. The solid was isolated by filtration, washed with cold THF (600 mL), and dried under vacuum to give fluoromethylcholine chloride as a white solid (1.14 kg, 70.7% yield, 98.0% purity). $^1$H-NMR (400 MHz, $D_2O$): 5.44 (s, 1H, -N-C$\underline{H}_2$-F), 5.32 (s, 1H, -N-C$\underline{H}_2$-F), 4.04–3.98 (m, 2H, -CH$_2$-C$\underline{H}_2$-OH), 3.60–3.54 (m, 2H, -N-C$\underline{H}_2$-CH$_2$-), 3.19 (s, 6H, -N(C$\underline{H}_3)_2$); $^{13}$C-NMR (100 MHz, $D_2O$): 97.8 and 95.6 (-N-$\underline{C}$H$_2$-F), 62.9 (-CH$_2$-$\underline{C}$H$_2$-OH), 55.1 (-N-$\underline{C}$H$_2$-CH$_2$-), 48.0 (-N($\underline{C}$H$_3)_2$); HRMS (ESI/TOF): m/z (M$^+$) calculated for $C_5H_{13}FNO$ (M$^+$) 122.0976, found 122.0975.

## Ethanol feeding trials in mice

All mice were maintained in an Association for the Assessment and Accreditation of Laboratory Animal Care, International-approved animal facility. All experimental protocols were approved by the Institutional Animal Care and Use Committee (IACUC) at the Cleveland Clinic. Age- and weight-matched female C57BL6/J mice were randomized into pair- and ethanol-fed groups and adapted to control liquid diet for 2 days. Two models of chronic ethanol feeding were used. (1) A 25-day chronic model in which mice were allowed free access to increasing concentrations of ethanol for 25 days (i.e. chronic feeding model) as previously described (*McCullough et al., 2018*). In this model, the ethanol-fed mice were acclimated to ethanol as follows: 1% vol/vol for 2 days, 2% vol/vol for 2 days, 4% vol/vol (22% kcal) for 1 week, 5% vol/vol (27% kcal) for 1 week, and last 6% vol/vol (32% kcal) for 1 week and is denoted as 32%, *day 25*. (2) A 10-day chronic model in which mice were allowed free access to a 5% vol/vol (27% kcal) for 10 days (*Bertola et al., 2013*). Ethanol-fed mice were allowed ad libitum access to liquid diet. Control mice were pair-fed a diet that received isocalorically substituted maltose dextrin for ethanol. Some cohorts received choline TMA lyase inhibitors IMC (0.06% wt/wt) or FMC (0.006% wt/wt) in these liquid diets throughout the entire 10- to 25-day feeding period. Lieber-DeCarli high-fat ethanol and control diets were purchased from Dyets (catalog number 710260; Bethlehem, PA).

## LPS injections

Female C57BL6/J mice at 10.5 weeks of age were injected intraperitoneally with either 15 mg/kg LPS (500 µg/mL, Sigma L4391) or a matched volume (30 mL/kg) of sterile saline. After 6 hr, mice were euthanized with ketamine/xylazine and the liver was immediately collected and homogenized in TRIzol. RNA was extracted using chloroform phase separation and purified using Qiagen RNeasy kit.

## Liver histology and immunohistochemistry

For histological analysis, formalin-fixed tissues were paraffin embedded, sectioned, and stained with hematoxylin and eosin. Formalin-fixed samples are coded at the time of collection for blinded analysis.

## Measurement of plasma aminotransferase levels

To determine the level of hepatic injury in mice, plasma was used to quantify ALT and AST levels using a commercially available enzymatic assay (Sekisui Diagnostics, Lexington, MA) according to manufacturer's instruction.

## Measurement of hepatic lipid levels

Extraction of liver lipids and quantification of total plasma and hepatic triglycerides, cholesterol, and cholesterol esters was conducted using enzymatic assays as described previously (*Warrier et al., 2015*; *Helsley et al., 2019*; *Schugar et al., 2017*; *Lord et al., 2016*).

## Quantification of TMA-related metabolites in acidified plasma

Stable isotope dilution high-performance liquid chromatography with on-line tandem mass spectrometry (LC-MS/MS) was used for quantification of levels of TMAO, TMA, choline, carnitine, betaine, and γ-butyrobetaine in plasma, as previously described (*Wang et al., 2014a*). Their d9(methyl) isotopologues were used as internal standards. LC-MS/MS analyses were performed on a Shimadzu 8050 triple quadrupole mass spectrometer. IMC and d2-IMC, along with other metabolites, were monitored using multiple reaction monitoring of precursor and characteristic product ions as follows: m/z 230.0 → 58.0 for IMC; m/z 232.0 → 60.1 for d2-IMC; m/z 76.0 → 58.1 for TMAO; m/z 85.0 → 66.2 for d9-TMAO; m/z 60.2 → 44.2 for TMA; m/z 69.0 → 49.1 for d9-TMA; m/z 104.0 → 60.1 for choline; m/z 113.1 → 69.2 for d9-choline; m/z 118.0 → 58.1 for betaine; m/z 127.0 → 66.2 for d9-betaine.

## Cecal microbiome analyses

16S rRNA amplicon sequencing were done for V4 region using via miSEQ from mouse cecal contents. Raw 16S amplicon sequence and metadata were *demultiplexed using split_libraries_fastq*.py script implemented *in QIIME1.9.1* (*Caporaso et al., 2010*). Demultiplexed fastq file was split into sample specific fastq files using split_sequence_file_on_sample_ids.py script from Qiime1.9.1 (*Caporaso et al., 2010*). Individual fastq files without non-biological nucleotides were processed using Divisive Amplicon Denoising Algorithm (DADA) pipeline (*Callahan et al., 2016*). The output of the dada2 pipeline (feature table of amplicon sequence variants [an ASV table]) was processed for alpha and beta diversity analysis using *phyloseq* (*McMurdie and Holmes, 2013*), and microbiomeSeq (http://www.github.com/umerijaz/microbiomeSeq) packages in R. Alpha diversity estimates were measured within group categories using estimate_richness function of the *phyloseq* package (*McMurdie and Holmes, 2013*). Multidimensional scaling (also known as principal coordinate analysis [PCoA]) was performed using Bray-Curtis dissimilarity matrix (*Knorr et al., 2020*) between groups and visualized by using *ggplot2* package (*Wickham, 2009*). We assessed the statistical significance ($p < 0.05$) throughout and whenever necessary, we adjusted p-values for multiple comparisons according to the Benjamini and Hochberg method to control false discovery rate (*Benjamini, 2010*) while performing multiple testing on taxa abundance according to sample categories. We performed an analysis of variance (ANOVA) among sample categories while measuring the of alpha diversity measures using plot_anova_diversity function in *microbiomeSeq* package (http://www.github.com/umerijaz/microbiomeSeq). Permutational multivariate analysis of variance (PERMANOVA) with 999 permutations was performed on all principal coordinates obtained during PCoA with the *ordination* function of the *microbiomeSeq* package. Wilcoxon (non-parametric) test was performed on ASV's abundances against metadata variables levels using their base functions in R (*Tilt, 1999*).

## RNA sequencing in mouse tissues

RNA sequencing libraries were generated from mouse liver using the Illumina mRNA TruSeq Directional library kit and sequenced using an Illumina HiSeq4000 (both according to the manufacturer's instructions). RNA sequencing was performed by the University of Chicago Genomics Facility, and data analysis and data availability are described in detail in the online supplement. Briefly, RNA samples were checked for quality and quantity using the Bio-analyzer (Agilent). RNA sequencing libraries were generated using the Illumina mRNA TruSEQ Directional library kit and sequenced using an Illumina HiSEQ4000 (both according to the manufacturer's instructions). RNA sequencing was performed by the University of Chicago Genomics Facility. Raw sequence files will be deposited in the Sequence Read Archive before publication (SRA). Single-end 100 bp reads were trimmed with Trim Galore (v.0.3.3, https://www.bioinformatics.babraham.ac.uk/projects/trim_galore/. ) and controlled for quality with FastQC (v0.11.3, http://www.bioinformatics.bbsrc.ac.uk/projects/fastqc) before alignment to the *Mus musculus* genome (Mm10 using UCSC transcript annotations downloaded July 2016). Reads were aligned using the STAR alignerSTAR in single pass mode (v.2.5.2a_modified, RRID:SCR_004463, https://github.com/alexdobin/STAR) with standard parameters. Raw counts were loaded into R (http://www.R-project.org/) and edgeR was used to perform upper quantile, between-lane normalization, and DE analysis. Values generated with the cpm function of edgeR, including library size normalization and log2 conversion, were used in figures. Heat maps were generated of top 50 differentially expressed transcripts using pheatmap. Reactome-based pathway analysis was performed using an open-sourced R package: ReactomePA. RNA sequencing data have been deposited into the National Institutes of Health (NIH)-sponsored GEO repository (accession number GSE157681).

## Phosphoproteomics analyses to examine TMA-induced signaling events in mouse liver

The goal of this experiment was to unbiasedly identify TMA-responsive signaling events in mouse liver after an acute exposure (10 min) of TMA. To closely mimic physiological route of delivery, we delivered saline or TMA directly into the portal vein in fasted mice. Briefly, C57BL/6 mice were fasted overnight (12 hr fast), and between the hours of 9:00–10:00 am (2–3 hr into light cycle), mice were anesthetized using isoflurane (4% for induction and 2% for maintenance). Once fully anesthetized, a midline laparotomy was performed, and the portal vein was visualized under a Leica M650 surgical microscope. Briefly, a fresh 10 mM stock of trimethylamine hydrochloride (TMA-HCL) made in sterile saline, and the

pH of stock solution was adjusted to 7.4. Mice then received 20 µL of either saline vehicle or TMA-HCL via direct syringe infusion (Becton-Dickson product #309306); 9.75 min later a small aliquot (50 µL) of portal blood was collected by pulling back on injection syringe left in place following injection. In saline vehicle injected mice, portal blood levels of TMA ranged from 0.49 to 2.22 µM and TMAO levels ranged from 2.53 to 7.14 µM. In mice injected with TMA-HCL, portal blood levels of TMA ranged from 125.36 to 319.55 µM and TMAO levels ranged from 9.68 to 17.48 µM. Exactly 10 min after initial injection, the liver was rapidly snap-frozen by immersion in liquid nitrogen. Liver samples were homogenized, the protein was precipitated with acetone, and the protein concentration was measured. A total of 1 mg of protein from each sample was digested with trypsin and the resulting tryptic peptides were subjected to phosphoserine and phosphothreonine enrichment using the Thermo Scientific Pierce $TiO_2$ Phosphopeptide Enrichment and Clean-up Kit (Fisher # PI88301). The enrichment was performed based on the manufacturer's instructions. The enriched peptide samples were subjected to C18 clean-up prior to LC-MS analysis. The LC-MS system was a Finnigan LTQ-Obitrap Elite hybrid mass spectrometer system. The HPLC column was a Dionex 15 cm × 75 µm id Acclaim Pepmap C18, 2 µm, 100 Å reversed-phase capillary chromatography column. Five µL volumes of the extract were injected and the peptides eluted from the column by an acetonitrile/0.1% formic acid gradient at a flow rate of 0.25 µL/min were introduced into the source of the mass spectrometer on-line. The microelectrospray ion source is operated at 1.9 kV. The digest was analyzed using the data-dependent multitask capability of the instrument acquiring full scan mass spectra to determine peptide molecular weights and product ion spectra to determine amino acid sequence in successive instrument scans. The LC-MS/MS data files were searched against the mouse UnitProtKB database (downloaded in December 2019 contains 17,017 sequences) using Sequest bundled into Proteome Discoverer 2.4. Cysteine carbamidomethylation was set as a fixed modification and oxidized methionine, protein N-terminal acetylation, and phosphorylation of serine, threonine, and tyrosine were considered as dynamic modification. A maximum of two missed cleavages were permitted. The peptide and protein false discovery rates were set to 0.01 using a target-decoy strategy. Phosphorylation sites were identified using ptmRS node in PD2.4. The relative abundance of the positively identified phosphopeptides was determined using the extracted ion intensities (Minora Feature Detection node) with Retention time alignment. All peptides were included in the quantitation, the peptide intensities were normalized to total peptide amount. Missing values were imputed in Perseus using a normal distribution. A total of 789 phosphopeptides were identified with 36 phosphopeptides determined to be two-fold different in the TMA and saline samples with a p-value < 0.05 (t-test).

## Statistical analysis

All statistical analyses were performed using GraphPad Prism and p < 0.05 was considered statistically significant. All data are presented as mean ± SEM, unless otherwise noted in the figure legends. All data were tested for equal variance and normality. For two-group comparison of parametric data, a two-tailed Student's t-test was performed, while non-parametric data were analyzed with Mann-Whitney U test (also called the Wilcoxon rank-sum test). For studies comparing vehicle and TMA lyase inhibitors in pair- and ethanol-fed mice, a two-way ANOVA was performed, followed by Tukey's tests for post hoc analysis. For human studies in AH patients, statistical significance was determined by ANOVA and a Tukey's honest significant difference post hoc test (p < 0.05).

## Acknowledgements

This work was supported in part by National Institutes of Health grants P50 AA024333 (AJM, SD, DSA, LEN, JMB), R01 DK120679 (JMB), P01 HL147823 (JMB, SLH), U01 AA026938 (LEN, JMB), P50 CA150964 (JMB), U01 AA021890 (LEN, SD), U01 AA021893 (SD, BB, CJM, MM, GS, and AJM), R01 HL103866 (SLH), R01 HL144651 (ZW), R01 HL130819 (ZW), U01 AA026980 (CJM), P50 AA 024337 (CJM), R21 AR 071046 (SD), R01 GM119174 (SD), R01 DK113196 (SD), R56 HL141744 (SD), U01 DK061732 (SD), U01 AA026977 (GS), UH3 AA026970 (GS), K99 AA028048 (AK), a Leducq Transatlantic Networks of Excellence Award (SLH), a JSPS Overseas Research Fellowship 201960331 (TM), and the American Heart Association (Postdoctoral Fellowships 17POST3285000 to RNH and 15POST2535000 to RCS). The Orbitrap Elite instrument used for proteomics was purchased via an NIH shared instrument grant 1S10RR031537 (BW).

# Additional information

### Competing interests

Zeneng Wang: Kaiser Permanente (CME lecture sessions) Advisory Board for Incyte (on treatment of cholangiocarcinoma). Stanley L Hazen: Z.W. report being named as co-inventor on pending and issued patents held by the Cleveland Clinic relating to cardiovascular diagnostics and therapeutics. Z.W. reports being eligible to receive royalty payments for inventions or discoveries related to cardio-vascular diagnostics or therapeutics from Zehna Therapeutics, Cleveland Heart Lab, a wholly owned subsidiary of Quest Diagnostics, and Procter & Gamble. The other authors declare that no competing interests exist.

### Funding

| Funder | Grant reference number | Author |
|---|---|---|
| National Institutes of Health | P50 AA024333 | Arthur J McCullough<br>Srinivasan Dasarathy<br>Daniela S Allende<br>Laura E Nagy<br>Jonathan Mark Brown |
| National Institutes of Health | R01 DK120679 | Jonathan Mark Brown |
| National Institutes of Health | P01 HL147823 | Jonathan Mark Brown<br>Stanley L Hazen |
| National Institutes of Health | U01 AA026938 | Laura E Nagy<br>Jonathan Mark Brown |
| National Institutes of Health | P50 CA150964 | Jonathan Mark Brown |
| National Institutes of Health | U01 AA021890 | Laura E Nagy<br>Srinivasan Dasarathy |
| National Institutes of Health | U01 AA021893 | Srinivasan Dasarathy<br>Bruce Barton<br>Craig J McClain<br>Marko Mrdjen<br>Gyongyi Szabo<br>Arthur J McCullough |
| National Institutes of Health | R01 HL103866 | Stanley L Hazen |
| National Institutes of Health | R01 HL144651 | Zeneng Wang |
| National Institutes of Health | R01 HL130819 | Zeneng Wang |
| National Institutes of Health | U01 AA026980 | Craig J McClain |
| National Institutes of Health | P50 AA 024337 | Craig J McClain |
| National Institutes of Health | R21 AR 071046 | Srinivasan Dasarathy |
| National Institutes of Health | R01 GM119174 | Srinivasan Dasarathy |
| National Institutes of Health | R01 DK113196 | Srinivasan Dasarathy |
| National Institutes of Health | R56 HL141744 | Srinivasan Dasarathy |
| National Institutes of Health | U01 DK061732 | Srinivasan Dasarathy |

| Funder | Grant reference number | Author |
|---|---|---|
| National Institutes of Health | U01 AA026977 | Gyongyi Szabo |
| National Institutes of Health | UH3 AA026970 | Gyongyi Szabo |
| National Institutes of Health | K99 AA028048 | Anagha Kadam |
| National Institutes of Health | 1S10RR031537 | Belinda Willard |
| National Institutes of Health | Leducq Transatlantic Networks of Excellence Award | Stanley L Hazen |
| JSPS Overseas Research Fellowship | 201960331 | Tatsunori Miyata |
| American Heart Association | 17POST3285000 | Robert N Helsley |
| American Heart Association | 15POST2535000 | Rebecca C Schugar |

The funders had no role in study design, data collection and interpretation, or the decision to submit the work for publication.

## Author contributions

Robert N Helsley, Chase Neumann, Lucas J Osborn, Rebecca C Schugar, Megan R McMullen, Annette Bellar, Kyle L Poulsen, Adam Kim, Vai Pathak, Marko Mrdjen, James T Anderson, Belinda Willard, Craig J McClain, Mack Mitchell, Arthur J McCullough, Svetlana Radaeva, Bruce Barton, Gyongyi Szabo, Srinivasan Dasarathy, Jose Carlos Garcia-Garcia, Daniel M Rotroff, Zeneng Wang, Stanley L Hazen, Laura E Nagy, Jonathan Mark Brown, Conceptualization, Data curation, Formal analysis, Funding acquisition, Investigation, Methodology, Project administration, Resources, Software, Supervision, Validation, Visualization, Writing – original draft, Writing – review and editing; Tatsunori Miyata, Conceptualization, Data curation, Formal analysis, Investigation, Methodology, Validation, Visualization, Writing – original draft, Writing – review and editing; Anagha Kadam, Data curation, Formal analysis, Investigation, Methodology, Writing – original draft, Writing – review and editing; Venkateshwari Varadharajan, Naseer Sangwan, Emily C Huang, Rakhee Banerjee, Amanda L Brown, Conceptualization, Data curation, Formal analysis, Investigation, Methodology, Writing – original draft, Writing – review and editing; Kevin K Fung, Conceptualization, Data curation, Formal analysis, Investigation, Methodology, Project administration, Supervision, Validation, Visualization, Writing – original draft, Writing – review and editing; William J Massey, Conceptualization, Data curation, Formal analysis, Funding acquisition, Investigation, Methodology, Project administration, Resources, Supervision, Validation, Visualization, Writing – original draft, Writing – review and editing; Danny Orabi, Conceptualization, Data curation, Formal analysis, Writing – original draft, Writing – review and editing; Daniela S Allende, Data curation, Formal analysis, Methodology, Visualization, Writing – original draft, Writing – review and editing

## Author ORCIDs

Robert N Helsley http://orcid.org/0000-0001-5000-3187
William J Massey http://orcid.org/0000-0002-2087-6048
Bruce Barton http://orcid.org/0000-0001-7878-8895
Srinivasan Dasarathy http://orcid.org/0000-0003-1774-0104
Jonathan Mark Brown http://orcid.org/0000-0003-2708-7487

## Ethics

Clinical trial registration NCT01809132; NCT03224949.
Human subjects: Patients with AH were classified as moderate (MELD < 20, n=112) and severe (MELD ≥20, n=152) according to the MELD score at admission as part of either of two independent clinical trials (ClincalTrials.gov identifier # NCT01809132 and NCT03224949) or the NOAC biorepository. These studies were approved by the Institutional Review Boards of all 4 participating institutions

and all study participants consented prior to collection of data and blood samples. Written informed consent was obtained from each patient included in the study and the study protocol conforms to the ethical guidelines of the 1975 Declaration of Helsinki as reflected in a priori approval by the Institutional Review Boards at Johns Hopkins Medical Institutions.

All mice were maintained in an Association for the Assessment and Accreditation of Laboratory Animal Care, International-approved animal facility. All experimental protocols were approved by the institutional animal care and use committee (IACUC) at the Cleveland Clinic.

### Decision letter and Author response

Author response https://doi.org/10.7554/eLife.76554.sa2

## Additional files

### Supplementary files
• Transparent reporting form

### Data availability

Sequencing data have been deposited in GEO under accession code GSE157681.

The following dataset was generated:

| Author(s) | Year | Dataset title | Dataset URL | Database and Identifier |
|---|---|---|---|---|
| Brown JM, Helsley R, Kadam A, Neumann C | 2021 | The Gut Microbe-Derived Metabolite Trimethylamine is a Biomarker of and Therapeutic Target in Alcohol-Associated Liver Disease | http://www.ncbi.nlm.nih.gov/geo/query/acc.cgi?acc=GSE157681 | NCBI Gene Expression Omnibus, GSE157681 |

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
