## [Editor Report]

This paper aims to understand the mechanisms by which gut microbes synergize with excessive alcohol intake to cause liver injury, and whether drugs that selectively target gut microbial metabolism can improve alcohol-associated liver disease (ALD). The authors used liquid chromatography tandem mass spectrometry to quantify the levels of microbe and host choline co-metabolites in controls and patients with alcohol-associated hepatitis (AH). They also treated mice with bacterial choline trimethylamine (TMA) lyase inhibitors to reduce gut microbe-dependent TMA production, followed by measurement of Indices of liver injury. They showed that gut microbial choline metabolite TMA is increased in AH patients, which correlates with reduced liver expression of the TMA oxygenase Flavin-containing monooxygenase 3 (FMO3). They also show that inhibition of gut microbial CutC/D activity protects from ethanol-induced liver injury in mouse models, which was associated with reorganization of the gut microbiome and host liver transcriptome. The authors conclude that microbial TMA is elevated in patients with AH, and inhibition of TMA production by gut microbes protects against ethanol-induced liver injury.

---

## [Author Response]

[Editors' note: we include below the reviews that the authors received from another journal, along with the authors’ responses.]

We have spent the last 9 months performing additional experiments to address reviewer concerns, including several additional mouse studies as well as increasing the number of patient samples under investigation. Here we provide a point-by-point response to each reviewer comment.

Reviewers' comments:Board of Editors:Interesting study, which is unfortunately too preliminary for some aspects of the work. Some additional analyses are needed based on the following major comments:1. Many mouse experiments have limited sample size (n=4 to 5) without replication. Increase in the sample size and/or independent replications are needed.

To address this valid point, we have conducted additional mouse experiments to increase the reproducibility of the results. As reviewer #2 astutely pointed out, there is an interaction with IMC + ethanol leading to reduced food intake (Supplemental Figure 2). Given this, we chose to focus the additional mouse experiments on the more potent TMA lyase inhibitor, FMC. Data shown in the original submission indicate that FMC treatment reduces ethanol-induced liver injury in a chronic model of ethanol feeding (Figure 3). In the revised manuscript, we have treated mice with the TMA lyase inhibitor FMC in a separate model of ethanol-induced liver injury. This has allowed us to both replicate findings in another model and also increase sample size. These data are included in Supplementary Figure 3.

2. Alcohol increases TMA in Figure 2 but not in Figure 3. Please justify this discrepancy. It might resolve, if the sample size is expanded to sufficient numbers.

We agree that there is a discrepancy between TMA and TMAO level across the different studies done. We suspect that this is in large part due to the timing of when the blood samples were collected, which is a critical determinant of circulating TMA and TMAO. We and others have previously reported (PMIDs: 29172946, 26265295, 26465927, 34405585, 23614584 ) that TMA and TMAO levels are highly dependent on the time of the last meal (i.e. when dietary substrates are presented to gut microbes). Furthermore, in another project we have under late stage revision at a preprint server (https://www.biorxiv.org/content/10.1101/2020.12.04.411546v1) we have found that the TMAO pathway is under complex circadian regulation and the highest levels are actually at night when mice are actively eating. Collectively, we have come to appreciate that plasma TMA and TMAO levels can be dramatically different based on the timing of the last meal as well under circadian regulation. However, it is important to note that the studies here were not designed to evaluate the peak levels of TMA and TMAO, and instead aimed to test the ability of TMA lyase inhibitors (IMC and FMC) to blunt the chronic effects of ethanol feeding. It is most common to perform necropsies for such ethanol feeding trials during the early morning to avoid circadian changes, so we felt it was important to stay consistent in our goal to focused on ethanol-induced liver injury phenotypes. Although the levels of TMA are not consistently altered across all studies in this manuscript, we do not feel this negatively impacts the primary outcome of the study that gut microbe-targeted TMA lyase inhibitors improve ethanol-induced liver injury. Regardless of whether ethanol either acutely or chronically elevates TMA levels, our data show that gut microbe-targeted TMA lyase inhibitors can provide some benefit in different mouse models of ethanol-induced liver injury which is an important advance.

3. Another discrepancy is the lack of difference in circulating TMA between moderate and severe alcoholic hepatitis while hepatic FMO3 is reduced in severe AH but not in "early AH". This suggests that a decreased FMO3 does not explain why TMA is high during AH.

The liver FMO3 mRNA levels in the early AH start to decrease compared to healthy controls, albeit not statistically significant. We suspect that the level of hepatic inflammation likely dictates the level of FMO3 expression. To directly test this hypothesis, we performed a new experiment in mice where we injected a high dose of the toll like receptor 4 (TLR4) agonist lipopolysaccharide (LPS) to induce strong hepatic inflammation, and see that Fmo3 mRNA expression is reduced by ~ 70% in mouse liver (Figure 1E). These new data show that activation of TLR4-dependent signaling, which is a hallmark of ethanol consumption, promotes reduction in Fmo3 expression. Serendipitously, in this same experiment we also found that the only known host TMA receptor (PMID: 23393561, 23177478) called trace-amine associated receptor 5 (TAAR5) was upregulated by LPS treatment. We agree with the possibility that decreased FMO3 may not be the only reason for increased TMA in alcohol-associated hepatitis patient or in mouse models of ethanol consumption, and it is very likely that broad changes in gut microbiome community structure (i.e. overgrowth of TMA producers) is also likely involved.

4. In line with the reviewers' comments, more experiments are needed to provide mechanistic insight (see comments by reviewer #1, more severe ALD models, FMO3-ko mice).

We appreciate these suggestions and have spent the past 9 months performing additional models of ethanol-induced liver injury using our lead TMA lyase inhibitor (FMC), and also performed ethanol feeding studies in mice genetically lacking Fmo3 (Fmo3-ko). Results from these studies are discussed below and/or included in the revised manuscript.

Reviewer #1:Several recent studies reported that breath levels of the primary metabolite TMA and other related cometabolites are elevated in ALD patients. In the current paper, the authors reported that serum levels of TMA not TMAO were elevated in patients with moderate or severe AH compared to healthy controls. By using chronic ethanol feeding model with the treatment of CutC/D inhibitors, the authors found that small molecule inhibition of gut microbial CutC/D activity protected mice from ethanol-induced liver injury. Finally, the authors performed cecal microbiome analyses (Fig, 4) and liver RNA seq analyses (Figure 5) from CutC/D inhibitor-treated ethanol-fed mice and phosphoproteomics analyses of TMA infused liver (Fig, 6).Although the authors confirmed elevation of TMA in patients with AH and in chronic ethanol-fed mice, the role of TMA was tested in chronic ethanol-fed mice with two inhibitors, the molecular mechanisms by which TMA contributes to the pathogenesis of AH are not clear. The important role of TMA in AH should be tested in more severe ALD models and in FMO3 knockdown or knockout mice.Specific Comments:1. The liver enzyme flavin-containing monooxygenase 3 (FMO3) is the predominant TMA to TMAO converting enzyme in the adult liver. In Figure 1C, the authors show that mRNA levels for FMO3 are uniquely repressed in patients with AH, but not in other liver disease etiologies such as NAFLD or viral hepatitis. However, are the expression levels of FMO3 related to the disease severity (such as the loss of hepatocytes)? In Figure 1C, the authors used AH with liver failure (AHL, n = 18), explant tissue from patients with severe AH; however, NAFLD, HCV, and HCV cirrhosis seem less severe compared to AH samples. It will be interested to test whether serum TMA levels are also elevated in patients with end-stages of NAFLD or HCV, and whether hepatic expression of FMO3 mRNA is decreased in these samples.

The reviewer raises a valid point about the loss of hepatocytes as the culprit behind the FMO3 expression levels across varying liver etiologies. However, this is a difficult thing to model in rodents as is the focus of this current study. We agree that it would be of interest to measure plasma TMA in the same patients described in Figure 1C (i.e. across different liver disease etiologies). However, the residual plasma samples from those patients were not properly acidified precluding us from being able to detect TMA. It is important to note that since our original submission several new papers have emerged showing that plasma TMAO levels are actually increased in subjects with non-alcoholic fatty liver disease (NAFLD) and are associated with non-alcoholic steatohepatitis (NASH) in obese subjects with type 2 diabetes (PMIDs: 32791310, 33993608). Interestingly, elevated levels of TMAO in NAFLD is associated with increase all-cause mortality (PMID: 33993608). Another recent study reported that circulating TMAO is reduced in subjects with advanced hepatitis B-related hepatocellular carcinoma (PMID: 27122669). Although beyond the scope of this body of work, these results indicate that additional validation cohorts are needed to determine whether the TMA-FMO3TMAO is specifically altered in different liver disease etiologies.

2. In Figure 1, serum TMA levels were much higher in patients with moderate or severe AH. How about serum TMA levels in alcoholics without liver diseases and alcoholic cirrhosis?

At this point we do not have access to acidified plasma or serum samples from alcoholics with liver disease and/or alcoholic cirrhosis. We are attempting to actively recruit such patients through our collaborative network, but feel this is beyond the scope of the current manuscript.

3. Hepatic expression of FMO3 is markedly downregulated in severe AH; How about hepatic expression of FMO3 in ethanol-fed mice in Figure 2, 3? It will be interesting to test whether knockdown or knockout of FMO3 elevates TMA and liver injury in mice after ethanol feeding.

We have spent substantial effort to address this extremely important issue, and provide here a summary of the findings. First, over 5 year ago we used an antisense oligonucleotide (ASO) knockdown approach that we have previously published (PMID: 25600868, 28636934) to reduce hepatic Fmo3 expression in ethanol-challenged mice. See Author response image 1, where there was a striking exacerbation of ethanol-induced inflammation and liver enzyme levels (ALT) in Fmo3 knockdown mice, yet an attenuation of ethanol-induced hepatic steatosis.

**Author response image 1. sa2fig1:** Inactivation of Flavin Monooxygenase 3 (FMO3)-Drive Conversion of Trimethylamine (TMA) to Trimethylamine-N-Oxide (TMAO) Exacerbates Ethanol-Induced Livery Injury. Female C57BL/6 mice were treated with a non-targeting control (Con) antisense oligonucleotide (ASO) or an ASO targeting the knockdown of FMO3 and subjected to the chronic (25 day) Lieber-DiCarli liquid diet feeding paradigm. (A) Hepatic FMO3 mRNA levels. (B) Ratio of FMO3’s substrate TMA to product in the circulation. (C) Plasma alanine aminotransferase A(LT) levels. (D) Hepatic Expression of monocyte chemoattractant protein 1 (MCP1). (E) Representative H&E stained liver sections (200x); arrows indicate localized immune cell infiltration. All data represent the mean + S.E.M. from 6 mice per group and means not sharing a common superscript differ (p<0.05).

Although these data are striking, we have decided not to include these data in the revised manuscript here because we have discovered and published on the

potential off target effects of Fmo3 ASOs (PMID: 31070450) and want to ensure high rigor and reproducibility in our published results.

After receiving the original reviewer comments in January 2021, as a more rigorous genetic approach we immediately bred littermate Fmo3+/+ and Fmo3-/- for ethanol feeding trials. This has been no small task to complete during the COVID-19 pandemic, but we were able to get 3 separate cohorts of mice through the chronic Lieber DiCarli ethanol feeding model. However, the results are somewhat inconclusive because in these studies there was a very modest effect of ethanol even in wild type (Fmo3+/+) mice (see Author response image 2). Although our line of Fmo3-/- mice has been backcrossed to C57BL/6J background, we suspect that in this current state of backcrossing that this mouse line is simply resistant to ethanol-induced liver injury (which we have seen to be very strain specific), which preclude strong interpretations. Although none of these results with Fmo3 loss-of-function satisfactorily address the question at hand, we wanted to be completely transparent of our attempts to be responsive to the reviewers. If you do feel these Fmo3 loss-of-function experiments are important to show in the manuscript, we are happy to include these data with clear discussion of the caveats.

**Author response image 2. sa2fig2:** Chronic ethanol feeding in Fmo3+/+ and Fmo3-/- mice. Female Fmo3+/+ and Fmo-/- mice were fed either ethanol-fed or pair-fed using the 25-day chronic feeding paradigm. Plasma levels of trimethylamine (TMA, trimethylamine N-oxide (TMAO)), choline, carnitine and betaine were measured by mass spectrometry. Plasma alanine aminotransferase (ALT), plasms aspartate aminotransferase (AST), liver triglycerides, and mRNA levels for several inflammatory cytokines were measured as in other studies. Statistics were completed by a two-way ANOVA followed by a Tukey’s multiple comparison test. *P<0.05; **P<0.01; ***P<0.001; ****P<0.0001. All data are presented as mean + S.E.M. from 4-6 mice per group.

4. The authors show that serum levels of TMA and TMAO are elevated in ethanol-fed mice in Figure 2A, B, but such elevation was not observed in Figure 3A, B. These results are inconsistent.

As described above, we fully acknowledge that there is a discrepancy between TMA and TMAO level across the different studies done. We suspect that this is in large part due to the timing of when the blood samples were collected, which is a critical determinant of circulating TMA and TMAO. We and others have previously reported (PMIDs: 29172946, 26265295, 26465927, 34405585, 23614584 ) that TMA and TMAO levels are highly dependent on the time of the last meal (i.e. when dietary substrates are presented to gut microbes). Furthermore, in another project we have under late stage revision at a preprint server (https://www.biorxiv.org/content/10.1101/2020.12.04.411546v1) we have found that the TMAO pathway is under complex circadian regulation and the highest levels are actually at night when mice are actively eating. Collectively, we have come to appreciate that plasma TMA and TMAO levels can be dramatically different based on the timing of the last meal and circadian regulation. However, it is important to note that the studies here were not designed to look at the peak levels of TMA and TMAO, and instead designed to test the ability of TMA lyase inhibitors (IMC and FMC) to blunt the chronic effects of ethanol feeding. It is most common to perform necropsies for such ethanol feeding trials during the early morning, so we felt it was important to stay consistent in our goal to focused on ethanol-induced liver injury phenotypes. Although the levels of TMA are not consistently altered across all studies in this manuscript, we do not feel this negatively impacts the primary outcome of the study that gut microbe-targeted TMA lyase inhibitors improve ethanol-induced liver injury. Regardless of whether ethanol either acutely or chronically elevates TMA levels, our data show that gut microbe-targeted TMA lyase inhibitors can provide some benefit in different mouse models of ethanolinduced liver injury which is an important advance.

5. Although the authors performed cecal microbiome analyses (Fig, 4) and liver RNA seq analyses (Figure 5) from CutC/D inhibitor-treated ethanol-fed mice, and phosphoproteomics analyses of TMA infused liver (Fig, 6), these results did not really give clear mechanisms how TMA contributes to the pathogeneses of ALD.

We agree that our current data only provide potential clues into mechanism, and additional studies will be required to fully understand the mechanism by which TMA contributes to the pathogenesis of ALD. Despite this, we feel strongly that the major impact of this manuscript stems from our seminal observation that TMA is elevated in AH patients, and that selective inhibition of the gut microbial enzyme that produces TMA from choline (CutC/D) protects mice from ethanol-induced liver injury. This is the first ever demonstration that a bacterially-targeted enzyme inhibitor can provide benefit in preclinical animal models of ALD, and bolsters the emerging concept that drugging microbial metabolism can potentially improve human health (PMID: 28389555).

Minor points:In abstract, line 27, "non-lethal non-lethal" is duplicate

This has been addressed in the revised abstract

Line 22 page 6, is "microbe-associated molecule patterns (MAMPs)" the same as " Pathogenassociated molecule patterns (PAMPs)"? if yes, it is better to use PAMPs.

We feel that MAMP is more appropriate here because PAMP also includes viral molecular patterns which is not the focus of this story, which is gut microbiome focused.

Reviewer #2:The aim of the paper was to understand how the gut microbioal TMA/TMAO pathway may contribute to alcoholic liver disease (ALD) susceptibility and progression.The paper and the subject are really interesting. The authors demonstrate that this pathway is involved in alcohol-fed mice and correlated the results in human ALD.1. The cohort of patients is not well described. Suppl Table 1 does not includes any clinical data (patients characteristics, alcohol consumption, liver function, liver enzymes…).

We have included a more detailed description of the study subjects in the revised Supplemental Tables 1 and 2.

2. Sometimes the authors use the words moderate/severe alcoholic hepatitis, and sometimes early/advanced/and liver failure. This is really unclear. What kind of patients are studied. Severe AH is not synonymous to advanced liver disease. Are several different cohorts of patients included? Are patients in Figure 1A and 1C similar of different. If yes, it must be explained. The number of patients is very low in Figure 1C.

We apologize for the lack of clarity and have removed any description of advanced liver disease. Instead, we have clearly indicated the subjects under study were patients with either moderate or severe alcoholic hepatitis.

3. Even if the “magic p< 0.05” is obtained in Figure 1A and 1B, the are many overlaps of values between healthy controls and alcoholic patients rendering unlikely the use of TMA and TMAO as marker of the disease. Moreover, what does it mean that TMA/TMAO is a marker? A marker of what? This overlap should be emphasized and discussed. The title, the section “what you need to know”, the discussion should be modified accordingly.

We agree that our study was not designed to establish TMA and TMAO as “biomarker” of alcoholic hepatitis, so the sections mentioned have been updated accordingly. We have removed any discussion of TMA or TMAO being a “marker” or “biomarker” throughout this manuscript.

4. Figure 2 and 3. Are control mice similar in these 2 figures? I don’t think so as the values of controls in the two figures are different. Therefore, how the authors explain twice the absence of Cyp2e1 induction in alcohol-fed mice in terms of mRNA (Figure 2N, 3N) and the clear induction of Cyp2e1 in terms of protein (Figure 2O, 3O). Accordingly, in FMC-treated mice, there is a slight increase of Cyp2e1 mRNA whereas the authors conclude to a blunted induction of CYP2E1 protein. I don’t understand these discrepancies. Figure 2M, what is the interpretation of TNFα decrease with treatment whereas alcohol intake does not increase its level ? Moreovere, there is no modification of TNFα in Figure 3M.

Ethanol feeding studies require mice to be pair-fed to control for caloric intake. Therefore, the control mice in Figures 2 and 3 are different. The pair-fed control mice in Figure 2 were used for the IMC experiment and the pair-fed control mice in Figure 3 were used for the FMC experiment. Upon further investigation into Cyp2e1 mRNA and CYP2E1 protein levels in additional mouse studies, we did not find a consistent difference in all studies so we have removed the Cyp2e1 data altogether. To further confirm that ethanol metabolism was not altered by our TMA lyase inhibitors, we also provide new data showing blood ethanol levels which were not altered by FMC treatment. Although less than satisfying, It is not uncommon to see variable ethanol-induced effects on cytokine levels (i.e. TNFa) in different ethanol-feeding trials in mice. We simply report the data for each study for full transparency.

5. There is a reduction of food intake in mice receiving IMC, therefore, all the results may be related to a decrease of alcohol consumption. Accordingly, food intake and weight gain/loss should be provided for FMC-treated mice. Is alcohol consumption and/or blood/urine concentration level similar between control and FMC-treated mice?

We do agree the protection with IMC is likely partly explained by the reduction in food intake. However, we feel it is important to include the IMC study data (in comparison to FMC effects) so that future research can focus on FMC or other downstream inhibitors that lack this potentially confounding effect. Food intake data (which directly reports on ethanol consumption) are included for all studies, and we have also now included blood alcohol levels for control versus FMC-treated mice which were similar.

6. Figure 6. Infusion of TMA into the portal circulation. No material/method are provided for this experiment. The authors write “we infused physiological levels of TMA directly into the portal circulation draining the gut (i.e. portal vein) of fasted mice”. What is physiological concentration of TMA in the portal vein in the model? In figures 2 and 3 plasma level are provided but not portal vein level. Which dose of TMA was infused? Is this dose similar to the physiological concentrations (in control and/or ethanol-fed) or higher? Does this injection modify lipid metabolism/genes as the major phenotype of alcohol-fed mice is liver steatosis ? I don’t understand the relationships between the reported modifications in this experiment and the phenotype of the mice.

We apologize for the limited details provided for this experiment, and apparent lack of clarity behind this study. The purpose of this experiment was to understand for the first time what signaling events occur acutely after TMA is sensed by the mouse liver. To address this we directly administered TMA via the portal vein (as it would be delivered from gut microbes) to unbiasedly characterize the acute (10 minute in vivo stimulation) signaling events in the liver. This study is important because there are essentially no data in the literature focused on defining TMA-induced signal transduction in the liver, other than the few papers describing that TMA can activate the G protein coupled receptor Trace amine-associated receptor 5 (TAAR5). Current published results only show that TMA can activate the Gs-α-coupled TAAR5 receptor using reporter systems (PMID: 23393561, 23177478). Our study is the first to characterize the acute effects of TMA on the global liver phosphoproteome, and provides the first clues into what major phosphorylation events occur acutely. These findings will allow downstream mechanistic investigation into TMA-induced signaling in future studies.

As described in the online supplement portal blood levels of TMA in normal fasted mice typically range from 0.49-2.2 mM (i.e. this is the baseline fasted level). Under our experimental conditions exactly 10 minutes after portal administration, we achieved a portal blood TMA range of 125.4-320.6 mM. Admittedly this is on the high end of physiological concentrations, we have seen TMA levels in the millimolar range in Fmo3-/- mice fed a high choline diet.

The detailed protocol description can be found in subsection “Phosphoproteomics Analyses to Examine TMA-Induced Signaling Events in Mouse Liver”

“The goal of this experiment was to unbiasedly identify TMA-responsive signaling events in mouse liver after an acute exposure (10 minutes) of TMA. […] Missing values were imputed in Perseus using a normal distribution. A total of 789 phosphopeptides were identified with 36 phosphopeptides determined to be two-fold different in the TMA and Saline samples with a p-value <0.05 (t-test).”

7. The Discussion section is too long and sometimes off topic.

We apologize for this; we have cut the discussion down to make it more clear and concise.

8. “What you need to know” section should be modified. The new findings cannot include TMA as a biomarker of advanced AH. 1- it has not been demonstrated; (2) “advanced” AH is not a disease; the name of the disease is “severe AH”. Impact/short summary section: too general; should focus on the results of the paper.

We apologize for this; all sections have been modified to reflect these suggestions. We have removed any discussion of “advanced AH” and have removed any indication of biomarker status.

Second decision letter and responses:

Response to Reviewer 2:Major comments:1. I have recommended to the author to clarify the definitions: alcoholic hepatitis / early / moderate / advanced. It has not been done in the whole manuscript: alcohol-associated hepatitis is not synonymous to severe alcoholic hepatitis: figure 1 C and its legend (early alcoholic hepatitis, alcoholic hepatitis with liver failure); lay summary… It really seems that these sections were not carefully reread.

We apologize for not completely responding to the reviewer’s suggestion in the revision. We have now carefully clarified the phenotypes of the patients with ALD used in our analysis. In part, some of the confusion may have from the fact that the available patient samples and datasets used slightly different enrollment criteria. We have clarified this and noted the different diagnostic criteria between data sets as a limitation to the study. We have included a revision of the supplemental Tables including more detailed description of the patients, and have updated the manuscript throughout to accurately reflect the patient diagnosis.

2. Supplemental Tables 1 and 2 are now written. The authors choose to define the severity alcoholic hepatitis according to the MELD score and not the Maddrey's discriminant function. Nevertheless, they obviously write (page 13 line 32) that Maddrey's discriminant function and Child-Pugh score are important. Therefore they should include these scores in supplemental tables 1 and 2. Moreover, in these 2 tables, the units of laboratory data must be given. The amount of alcohol consumption should be included.

These are now included as the reviewer requests. We have also included MDF and Child-Pugh, when available. Alcohol intake has been reported via an AUDIT questionnaire, which is an alcohol use disorders identification test; this has been provided in both supplemental tables. We had originally focused on MELD score in this Table, as that was the primary diagnostic indicator of disease severity used in the DASH consortium.

3. In this paper, the authors demonstrate in a mouse model that they can prevent alcoholinduced liver lesions. Therefore, in the "what you need to know – new findings", they should not write that production of TMA is a therapeutic target in advanced alcoholic hepatitis. It's a shortcut to the results: (i) again, in this section, I don't know what is "advanced" alcoholic hepatitis; (ii) the author describes a pathophysiological mechanisms and he should not write in this section of the manuscript that TMA production is a therapeutic target in patients with severe alcoholic hepatitis; (iii) The title of the manuscript should be changed accordingly.

We thank the reviewer for reminding us as to the purpose of the “new findings” section. We removed the “What you need to know” section and repurposed the “Highlights” section. Based off the reviewers’ suggestions, we have edited this to better reflect our actual results, rather than our interpretation of those results. For the additional comments:

i) The phrase Advanced hepatitis has been replaced with alcohol-associated hepatitis

ii) The “therapeutic target” hypothesis has been removed

iii) The title of the manuscript has been changed to “Gut Microbe-Derived Trimethylamine is Elevated in Alcohol-Associated Hepatitis and Contributes to Ethanol-Induced Liver Injury,” in order to better reflect the results of our studies.

4. TMA infusion in the portal vein. The "material and method" section is now written and included in the manuscript. Again, I am not sure of the relevance of such a way of administration. I don't understand the explanation of the authors who assert that they infused a physiological levels of TMA (page 18 line 32). I even think that the doses may be supraphysiological. The authors should be more clear and discuss the limitations of their results according to the dose of TMA and the way of administration. I think that giving a diet enriched in choline would have been more physiological than a direct infusion of TMA into the portal vein.

We apologize for the confusion in regards to this experiment. However, we do feel strongly that this experiment provides new important information for the field, and should be included in this body of work. The rationale for giving TMA directly into the portal vein is to mimic its delivery to the liver from the intestine. Given the fact that TMA smell like rotting fish it is nearly impossible to provide it in the diet (i.e. mice will not eat the diet). Furthermore, we also do not like the approach of peripheral administration (i.e. intraperitoneal or intravascular injection) because naturally TMA is made in the gut solely by bacteria and then enters the liver via the portal tract; making these peripheral approaches much less physiologic. We do agree with the reviewer however that this experiment is slightly above physiological levels, but by design we were trying to raise levels in vivo to very high levels to maximize responses. Given this key limitation, we have removed any mention of “physiological levels” and have included the following sentences in the results regarding the limitations of this experiment:

“It is important to note that in this experiment we provided very high levels of exogenous TMA via direct injection, and future studies should focus on more physiologically-relevant modes of TMA production like provision of gut bacteria that can naturally or be genetically engineered to produce high levels.”

The reviewer mentions that “a diet enriched in choline would have been more physiological than a direct infusion of TMA into the portal vein.” Although we agree that dietary choline supplementation can raise TMA and TMAO levels in animals, we respectfully disagree that this would provide relevant information to TMA-specific signal transduction like our experiment sets out to do. Provision of supplemental dietary choline impacts many downstream metabolites originating from bacteria (TMA and acetaldehyde) and the host (free choline, many species of phosphatidylcholine, acetylcholine) and also influences cholinergic signaling. If we simply provide choline, it is much more difficult to interrogate the specific signaling elicited by TMA. Although beyond the scope of this current study, we have designed a set of gut microbial communities that can or cannot make TMA, via genetic deletion of the gut microbial choline TMA lyase CutC. We have plans in the near future to follow up on the portal infusion studies to validate some the findings using this approach and others as well.

5. Graphical abstract / suppl Figure 5. the title "metaorgansim metabolism model" must be changed. A metaorganism model should include the other pathways involved in AH than the only TMA/TMAO pathway. Many other metabolites can be involved in ALD pathophysiology. The title should refer to the manuscript itself**.**

We have changed the headings/title of the graphical abstract to specifically refer to TMA/TMAO pathway rather than the broader term of “metaorganismal model”. The new graphical abstract now refers specifically the metaorganismal TMAO pathway which is the focus of this body of work.

6. The authors suggest that TMA and not TMAO is involved in the pathophysiology of ALD. Interestingly, it has been suggested in other papers that TMAO is involved in the pathophysiology of NAFLD/NASH. How can the authors reconcile these discrepancies?

We agree there are several reports that have shown an association between plasma TMAO and NAFLD/NASH (32791310, 26743949), but these are simply associations. To our knowledge there are no direct mechanistic links between TMAO and NAFLD/NASH. Given this manuscript is focused solely on alcohol-associated liver disease, we have focused our conclusion and discussion around ethanol-induced liver injury.

Third decision letter and responses:

The revised version of your manuscript entitled "Gut Microbial Trimethylamine is Elevated in Alcohol Associated Hepatitis and Contributes to Ethanol-Induced Liver Injury in Mice " has been evaluated by the original reviewers and by the board of editors and we are sorry to say that we still see some significant issues that need to be addressed.You will find the comments of Reviewer 2 below, who requests some changes in the graphical abstract and in Suppl Figure 5c.In addition and more importantly, there is a global consensus between reviewers and the board of editors to consider that the interpretation of Figure 6 is challenging.We feel that the final paper would be much improved by the addition of relatively simple experiments surrounding the conclusion of Figure 6:–Adding an active control (i.e. another microbial metabolite) to confirm the specificity of the results obtained with TMA–Adding additional doses of TMA to confirm the dose-dependent changes observed in the liver

1. We have indeed performed identical phosphoproteomic studies with other gut microbegenerated metabolites including a phenol acid called 4-hydroxyphenylacetic acid (4-HPAA). I have attached here this manuscript we have recently prepared for 4-HPAA in the context of NAFLD (currently under peer review). Included in this resubmission, we have included the manuscript (Osborn. et al. 2021) which is currently under revision at Cell Host Microbe. This manuscript uses a similar in vivo phosphoproteomic approach in Figure 5A-5C, and as you would expect uncovered many different pathways that are not overlapping with figure 6 here with TMA. Given this work is an integral part of another manuscript, it cannot be included here. However, the data does address the specificity question in your decision letter below.

2. Alternatively, if the editorial team thinks that the phosphoproteomic data provided in Figure 6 are hard to interpret, we would be happy to just remove that figure. However, we do think the data are useful for providing new leads to follow up on, but years of additional work will be needed to identify the TMA receptor systems and downstream signal transduction.

3. Unfortunately, we will not be able to perform the dose-response for in vivo phosphoproteomics as requested by the editors. Although we agree this would provide some additional information, it would be incremental in nature and would not change any interpretation of the current work. Given a backlog on our proteomics mass spectrometer and the holiday season we would not be able to finish those analyses until at least March of 2022, based on our current schedule. This timeline is simply not feasible for us with competitors manuscripts advancing to our stage, so these data would not be included in a revised submission. It is important for us to be first with this story so we will need to make a decision on publication route in the coming weeks.

Reviewer #2:The manuscript has been improved.In his answer, the author writes that giving choline to induce an elevation of TMA would not be specific. This is correct. In this case, if the author thinks that giving choline is irrelevant, the graphical abstract and suppl Figure 5c which suggest that the intake of choline, the consumption of meat, eggs or cheese worsens alcoholic liver disease should be changed. This is not demonstrated at all and is a shortcut that may mislead the reader.

We have now changed the graphical abstract to remove any food items. This revised figure has now been uploaded.